# Neuroevolution is a Competitive Alternative to Reinforcement Learning for Skill Discovery

**Felix Chalumeau**[*1]    **Raphael Boige**[*1]    **Bryan Lim**[2]    **Valentin Macé**[1]
**Maxime Allard**[2]    **Arthur Flajolet**[1]    **Antoine Cully**[†2]    **Thomas Pierrot**[†1]
[1]InstaDeep [2]Imperial College
{f.chalumeau, r.boige, v.mace, a.flajolet, t.pierrot}@instadeep.com
{bwl116, m.allard20, a.cully}@ic.ac.uk

## Abstract

Deep Reinforcement Learning (RL) has emerged as a powerful paradigm for training neural policies to solve complex control tasks. However, these policies tend to be overfit to the exact specifications of the task and environment they were trained on, and thus do not perform well when conditions deviate slightly or when composed hierarchically to solve even more complex tasks. Recent work has shown that training a mixture of policies, as opposed to a single one, that are driven to explore different regions of the state-action space can address this shortcoming by generating a diverse set of behaviors, referred to as skills, that can be collectively used to great effect in adaptation tasks or for hierarchical planning. This is typically realized by including a diversity term - often derived from information theory - in the objective function optimized by RL. However these approaches often require careful hyperparameter tuning to be effective. In this work, we demonstrate that less widely-used neuroevolution methods, specifically Quality Diversity (QD), are a competitive alternative to information-theory-augmented RL for skill discovery. Through an extensive empirical evaluation comparing eight state-of-the-art algorithms (four flagship algorithms from each line of work) on the basis of (i) metrics directly evaluating the skills' diversity, (ii) the skills' performance on adaptation tasks, and (iii) the skills' performance when used as primitives for hierarchical planning; QD methods are found to provide equal, and sometimes improved, performance whilst being less sensitive to hyperparameters and more scalable. As no single method is found to provide near-optimal performance across all environments, there is a rich scope for further research which we support by proposing future directions and providing optimized open-source implementations.

## 1 Introduction

In the past decade, Reinforcement Learning (RL) has shown great promise at tackling sequential decision making problems in a generic fashion, leading to breakthroughs in many fields such as games (Silver et al., 2017), robotics (Andrychowicz et al., 2020), and control in industrial settings (Degrave et al., 2022). However, neural policies trained with RL algorithms tend to be over-tuned to the exact specifications of the tasks and environments they were trained on. Even minor disturbances to the environment, to the starting state, or to the task definition can incur a significant loss of performance (Kumar et al., 2020; Pinto et al., 2017). A standard approach to improve generalization is to introduce more variations during training (Tobin et al., 2017) but this assumes we can foresee all possibilities, which is not always true in the real world. Even in settings where this is feasible, introducing a wide spectrum of variations will make the problem harder to solve and the resulting policy may not perform as well in the nominal case. Another approach consists in co-training an adversarial agent whose task is to perturb the environments so as to minimize the policy's performance (Pinto et al., 2017). However, adversarial methods are notoriously unstable in Deep Learning (Arjovsky & Bottou, 2017) and can also compromise performance in the nominal scenario.

---

[*]Equal Contribution
[†]Equal Supervision

To improve robustness without explicitly identifying all possible variations, jointly training multiple policies to solve the same task in diverse ways has emerged as a promising line of work in the RL literature (Kumar et al., 2020). To motivate the approach, consider the problem of learning a policy to control the joints of a legged robot with the goal of running as fast as possible. Any damage to the robot legs might affect an optimal policy's ability to make the robot run fast, if at all. Yet, many of the slightly sub-optimal policies to the original problem (e.g. a policy making the robot hop using only one leg) would perform equally well in this perturbed setting. Two seemingly-opposed main lines of work have been pursued to maximize both performance and diversity in a collection of policies.

RL-rooted approaches (Eysenbach et al., 2019; Sharma et al., 2019; Kumar et al., 2020) introduce a randomly-generated latent variable and parametrize the policy to be a function of the state as well as this latent variable. At training time, the latent variable is drawn from a static distribution and fed as an input alongside the state to the policy, effectively defining a mixture of policies. To encourage diversity among these policies, a term derived from information theory that depends both on the policy parameters and the latent variable is added to the objective function (hereinafter referred to as fitness function). This term is typically formulated as the mutual information between the latent variable and a subset of the policy's trajectory, possibly conditioned on observations from the past.

Neuroevolution-rooted approaches instead stem from the subfield of Quality Diversity (QD) optimization (Pugh et al., 2016; Cully & Demiris, 2017; Chatzilygeroudis et al., 2021) and combine the tools developed in this space with RL algorithms to get the best of both worlds (Nilsson & Cully, 2021; Pierrot et al., 2022). QD optimization aims at generating and maintaining large and diverse collections of solutions, as opposed to a single optimal solution in Optimization Theory, by imitating the natural evolution of individuals competing for resources in their respective niches. In comparison to traditional Evolutionary Strategies, QD algorithms explicitly use a mapping from solution to a vector space, referred to as behavior descriptor space, to characterize solutions and maintain a data structure, a *repertoire*, filled with high-performing solutions that cover this space as much as possible.

Evolutionary Strategies (possibly hybridized with RL algorithms) have proven to be a competitive alternative to RL algorithms for many common sequential-decision making problems (Pierrot et al., 2022; Salimans et al., 2017). Hence, it is natural to believe that QD algorithms could also be competitive with information-theory-augmented RL approaches to generate diverse populations of high-performing policies in similar settings. Yet, QD approaches remain neglected in skill-discovery studies (Kumar et al., 2020), perhaps because they lack the sample-efficiency of state-of-the-art RL algorithms, sometimes requiring two orders of magnitude more interactions with the environment to solve a task (Pierrot et al., 2022). While this is a significant shortcoming for real-world applications that cannot be accurately described by a computational model, simulators are readily available for many applications. Additionally, when the simulator and the algorithm are implemented using modern vectorized frameworks such as JAX (Bradbury et al., 2018) and BRAX (Freeman et al., 2021), evolutionary approaches are competitive with RL approaches in terms of total training time on an accelerator in spite of the low sample-efficiency of these methods (Lim et al., 2022).

Our contributions are the following. **(1.)** We provide extensive experimental evidence that QD methods are competitive with RL ones for skill discovery in terms of performance given fixed compute and training time budgets and hyperparameter sensitivity. Specifically, using environments taken from the QD and RL literature, we compare eight state-of-the-art skill-discovery methods from the RL and QD world on the basis of (i) metrics directly evaluating the skills' diversity, (ii) the skills' performance on adaptation tasks, and (iii) the skills' performance when used as primitives for hierarchical planning. **(2.)** We open source efficient implementations of all environments and algorithms[1] based on the QDax library[2]. Armed with these, running any of the experiments, some of which require hundreds of millions of environments steps, takes only 2 hours on a single affordable accelerator. **(3.)** We provide a detailed analysis of the strengths and weaknesses of all methods, we show that no single method outperforms all others on all environments, and we identify future research directions.

## 2 PRELIMINARIES AND PROBLEM STATEMENT

We consider sequential decision making problems formulated as Markov Decision Processes (MDPs) and defined by $(\mathcal{S}, \mathcal{A}, \mathcal{R}, \mathcal{P}, \gamma)$, where $\mathcal{S}$ is the state space, $\mathcal{A}$ is the action space, $\gamma \in [0, 1]$ is the discount factor, $\mathcal{R} : \mathcal{S} \times \mathcal{A} \to \mathbb{R}$ is the reward signal and $\mathcal{P} : \mathcal{S} \times \mathcal{A} \to \mathcal{S}$ is the transition function.

---

[1] https://github.com/instadeepai/qd-skill-discovery-benchmark
[2] https://github.com/adaptive-intelligent-robotics/QDax

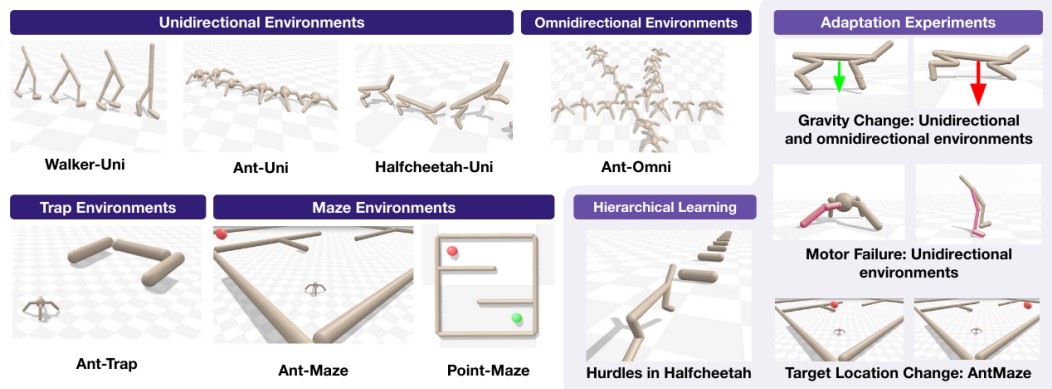

Figure 1: Illustrations of environments used for benchmarking (left) and perturbations applied to them for adaptation and hierarchical learning experiments (right). In UNIDIRECTIONAL environments, policies are trained to make the robots run forward with diverse gaits. In the OMNIDIRECTIONAL environment, the goal is to make a legged robot move on the 2D plane while minimizing the control energy. In ANT-TRAP, a robot learns to run in the forward direction while avoiding a trap. In ANT-MAZE and POINT-MAZE, a robot must reach a target position in a maze. Once trained over these tasks, policies are evaluated on perturbed versions of the tasks and environments.

Policies are assumed to be implemented by neural networks. We consider two cases. **(1.)** *A mixture of policies is conditioned on a latent variable $z \in \mathcal{Z}$,* in which case it is denoted by $\pi_\theta : \mathcal{S} \times \mathcal{Z} \to \mathcal{D}(\mathcal{A})$ where $\theta$ denotes the neural network weights and $\mathcal{D}(\mathcal{A})$ is the space of distributions over the action space. The collection of policies $\Pi$ is obtained by enumerating all possible values for the latent variable. **(2.)** *Policies share the same neural network architecture but have their own sets of parameters.* In this case, $\Pi$ corresponds to the collection of $N$ policies $\pi_{\theta_i} : \mathcal{S} \to \mathcal{D}(\mathcal{A})$, $i \in [1, N]$. We denote the trajectory of a single policy $\pi$ in the environment by $\tau \in \Omega$. The expected sum of rewards of a policy $\pi$, referred to as its fitness, is $F(\pi) = \mathbb{E}_\tau \sum_t \gamma^t \mathcal{R}(s_t, a_t)$.

In this work, compared to traditional RL, we consider a second objective: the policies in the collection $\Pi$, referred to as *skills*, must not only be high-performing but also diverse as a whole. The notion of diversity can be defined in various ways. In the RL literature, diversity is often measured by looking at the volume spanned by components of the trajectories induced by the policies, such as the sequences of visited states (Parker-Holder et al., 2020). In the QD literature, diversity is measured in a so-called *behavior descriptor space* $\Phi$ using a mapping $\phi : \Omega \to \Phi$ that is either hand-engineered or progressively refined in an unsupervised fashion. This approach is more general as it includes as a special case the first one but also allows to leverage expert knowledge on the task at hand, which is often available. We use the notion of behavior descriptor space in order to quantitatively compare the diversity of policies generated by RL and QD methods. Specifically, we define a bounded behavior descriptor space for each MDP, that may differ from the ones used internally by QD methods, which is discretized into a discrete tessellation of cells. Policies from $\Pi$ are inserted into the tessellation as a function of their behavior descriptors which enables us to use diversity metrics developed in the QD literature. This process is made precise in Section 5.2.

## 3    DIFFERENT SOLUTIONS TO THE SAME PROBLEM

**Information-theory-augmented RL methods.** Most methods aiming to maximize diversity in the RL literature, such as DIAYN (Eysenbach et al., 2019) and DADS (Sharma et al., 2019), fall into the class of latent-conditioned policies introduced in Section 2. They strive to maximize the Mutual Information (MI) between a latent variable $z$, which is drawn randomly from a pre-specified distribution, and some components of the trajectories induced by the conditioned policy. The MI maximization is approximately carried out through the introduction of a discriminator function $z \mapsto q(z|s)$ given a state $s \in \mathcal{S}$ (resp. a dynamics function $z \mapsto q(s_{t+1}|s_t, z)$ given the current state $s_t$ and the next state $s_{t+1}$) for DIAYN (resp. DADS) trained separately to differentiate the latent variables that lead to visit different states. The latent-conditioned policy is optimized using an off-policy model-free RL algorithm by taking the reward as the discriminator value, referred to as *intrinsic reward*.

DIAYN and DADS are unsupervised algorithms as they do not aim to maximize a reward function associated with a specific task. Recent work in MI-based RL, namely SMERL (Kumar et al., 2020), has extended DIAYN and DADS to supervised settings where both diversity and performance w.r.t. a reward function should be optimized. They formulate the problem as a Constrained MDP (CMDP) where the objective is to maximize the diversity of a mixture of policies, while also constraining the learned policies to be close to optimality w.r.t. the reward function. By solving these CMDPs, they obtain a collection of policies that are both diverse and high-performing. SMERL can use either DIAYN or DADS internally to generate collections of policies, two variants we refer to as SMERL(DIAYN) and SMERL(DADS). An alternative approach to SMERL simply consists in summing the task reward and the intrinsic reward and using the original DIAYN and DADS algorithms (Kumar et al., 2020; Osa et al., 2022; Gaya et al., 2021). This approach is very similar to Hausman et al. (2018). We refer to these variants as DIAYN+REWARD and DADS+REWARD. For clarity's sake, we only report the performance of the original DIAYN (resp. DADS) algorithm in the Appendix as it turns out to be significantly outperformed by DIAYN+REWARD (resp. DADS+REWARD) in our experiments.

**QD methods** fall into the second category of policies introduced in Section 2. These methods explicitly evolve and maintain a discrete repertoire of independent policies. QD algorithms traditionally rely on Evolutionary algorithms to incrementally update policies almost independently. Such techniques have the advantage of being easily parallelizable and can be used with non-differentiable policies and controllers (Tang et al., 2020). They have been shown to be viable alternatives to RL algorithms (Salimans et al., 2017).

MAP-ELITES is one of the most widespread QD algorithms. This method assumes that the behavior descriptor space has been discretized into a finite tessellation of cells, which together define a *repertoire* where policies are stored. This algorithm proceeds by iterations. At each iteration, a batch of policies is sampled uniformly from the repertoire and copied. Mutations and crossover operations are applied to them to obtain a new batch of policies which are evaluated in the environment to compute their fitnesses and behavior descriptors. For any of these policies, if the cell in the repertoire corresponding to its behavior descriptor is empty, then it is copied into this cell. Otherwise, the new policy replaces the current incumbent only if its fitness is higher than the current incumbent's fitness.

The exact choice of mutation and crossover operators significantly affects the performance of QD algorithms. Choices that rely exclusively on random perturbations typically require a significant number of iterations. To remedy this problem, several works have combined first-order optimization methods with Evolutionary algorithms to guide the parameter search (Colas et al., 2020; Pierrot et al., 2022). In particular, PGA-MAP-ELITES (Nilsson & Cully, 2021) is a hybrid algorithm that builds on the MAP-ELITES framework to select, mutate, and replace policies in the repertoire but further introduces a policy-gradient mutation operator which is used alongside the genetic operator. As part of every MAP-ELITES iteration, PGA-MAP-ELITES collects the experience of all evaluated policies and stores it into a replay buffer in order to train a pair of critics using the TD3 algorithm (Fujimoto et al., 2018). These critics are used to compute the policy-gradient updates applied to the policies.

While MAP-ELITES and PGA-MAP-ELITES are provided with a pre-specified behavior descriptor space, it is defined in an unsupervised fashion in a recent extension to MAP-ELITES dubbed AURORA (Cully, 2019; Grillotti & Cully, 2022a). This is achieved by training an autoencoder to reconstruct the trajectories collected in the environments and defining the behavior space as the latent space of the autoencoder. In this paper, we introduce a new unsupervised method which extends both AURORA and PGA-MAP-ELITES. This method, referred to as PGA-AURORA, use the same policy-gradient mutation operator as PGA-MAP-ELITES and the same unsupervised technique as AURORA to define the behavior descriptor space.

## 4 RELATED WORK

Historically, QD algorithms originated from Evolutionary algorithms, which also implement selection, mutation, and evaluation of populations of solutions. Novelty Search (NS) (Lehman & Stanley, 2011a) first introduced the idea of searching for novelty alone. In a sense, NS is similar to DIAYN and DADS as it is an exploration-only algorithm. Novelty Search with Local Competition (NSLC) (Lehman & Stanley, 2011b) and MAP-ELITES (Mouret & Clune, 2015) are two of the most popular QD algorithms. NSLC builds off the NS algorithm and, in contrast to MAP-ELITES, it maintains an unstructured repertoire of solutions that are selected for their performance when compared to similar

individuals. Relying on the same diversity-seeking strategies, QD-ES algorithms, which combine QD and Evolutionary Strategies (ES), such as NSR-ES and NSRA-ES, have been successfully applied to challenging continuous control environments (Conti et al., 2018). Aiming for improved sample efficiency, Colas et al. (2020) developed ME-ES which optimizes both for quality and diversity using MAP-ELITES and two ES populations. Similarly, CMA-ME (Fontaine et al., 2020) uses a covariance matrix adaptation technique to model and update a population distribution to sample individuals from and also maintains a population of varied emitters to generate solutions. More recently, some authors proposed to incorporate a policy-gradient component in the evolutionary framework (Nilsson & Cully, 2021; Pierrot et al., 2022; Tjanaka et al., 2022). Inspired by PGA-MAP-ELITES, the authors of QD-PG (Pierrot et al., 2022) introduced a diversity policy gradient to make diversity information available at the timestep level.

Multiple methods from the RL literature build an intrinsic reward with MI to learn a diverse set of skills. They are similar in spirit to DIAYN and can be adapted, along the same lines as what is done for SMERL, to use extrinsic rewards. Gregor et al. (2016) originally introduced MI for skill discovery. Most recent efforts in the skill-discovery RL literature focus on ways to extend DIAYN and DADS to improve the state coverage of the learned skills by using contrastive learning (Laskin et al., 2022), adding constraints to the skills (Hansen et al., 2021; Kamienny et al., 2021), or changing the loss function (Durugkar et al., 2021). Nevertheless, Zahavy et al. (2021) introduced an original approach where the intrinsic rewards are defined based on successors features rather than MI. We choose DIAYN, DADS, and SMERL as baselines in our experiments as they are the most widely used RL methods for skill discovery at the time of writing. There also exist methods based on tools other than MI that strive to evolve a diverse population of RL agents, such as Zhou et al. (2022); Parker-Holder et al. (2020); Zhang et al. (2019); Masood & Doshi-Velez (2019), but these studies mostly consider diversity as a means to increase performance w.r.t. the rewards, whereas DIAYN, SMERL, and DADS are evaluated on pure diversity, on adaptation tasks, or in hierarchical learning settings.

## 5 EXPERIMENTS

In this section, we present the frameworks used to compare the methods listed in Section 3 and analyze the results. First, in Section 5.1, we introduce the environments and tasks upon which the frameworks are built. In Section 5.2, we leverage tools and metrics from the QD literature to directly evaluate the diversity of the solutions found by each method in the process of solving the tasks. In Section 5.3, we investigate how diversity translates into robustness with few-shot adaptation experiments inspired from the RL literature. Finally, in Section 5.4, we evaluate the policies found by each method when used as primitives in a hierarchical learning setting.

### 5.1 ENVIRONMENTS

In order not to favor any method in the study, we use three different categories of environments and tasks: (i) continuous control locomotion tasks from the RL literature, (ii) exploration tasks with deceptive rewards from the QD literature, and (iii) hybrid tasks mixing both types as a middle ground. All environments are illustrated on Figure 1.

**Low-dimensional exploration task with deceptive rewards.** Following several works from both the skill-discovery RL and QD communities (Eysenbach et al., 2019; Kamienny et al., 2021; Campos et al., 2020), we consider a simple navigation environment where a point is controlled to navigate on a 2D plane to a specified target position. To turn the task into a challenging exploration problem, we introduce a maze through which the point must navigate to reach the target position. We refer to this task as POINT-MAZE.

**High-dimensional locomotion tasks.** We consider three locomotion tasks, namely ANT, HALFCHEE-TAH and WALKER, often used in the RL community to evaluate algorithms, including in RL works focused on diversity-seeking strategies (Nilsson & Cully, 2021; Eysenbach et al., 2019; Fujimoto et al., 2018). In these environments, the problem is to move legged robots by applying torques to their joints through actuators. These tasks are challenging for evolutionary algorithms: they typically require orders of magnitude more environment interactions than RL methods to evolve competitive policies (Colas et al., 2020; Pierrot et al., 2022). We define two subsets of tasks for these environments. The first subset, referred to as UNIDIRECTIONAL locomotion tasks and with the reward signal defined as the x-axis velocity minus the energy consumption, includes ANT-UNI, WALKER-UNI,

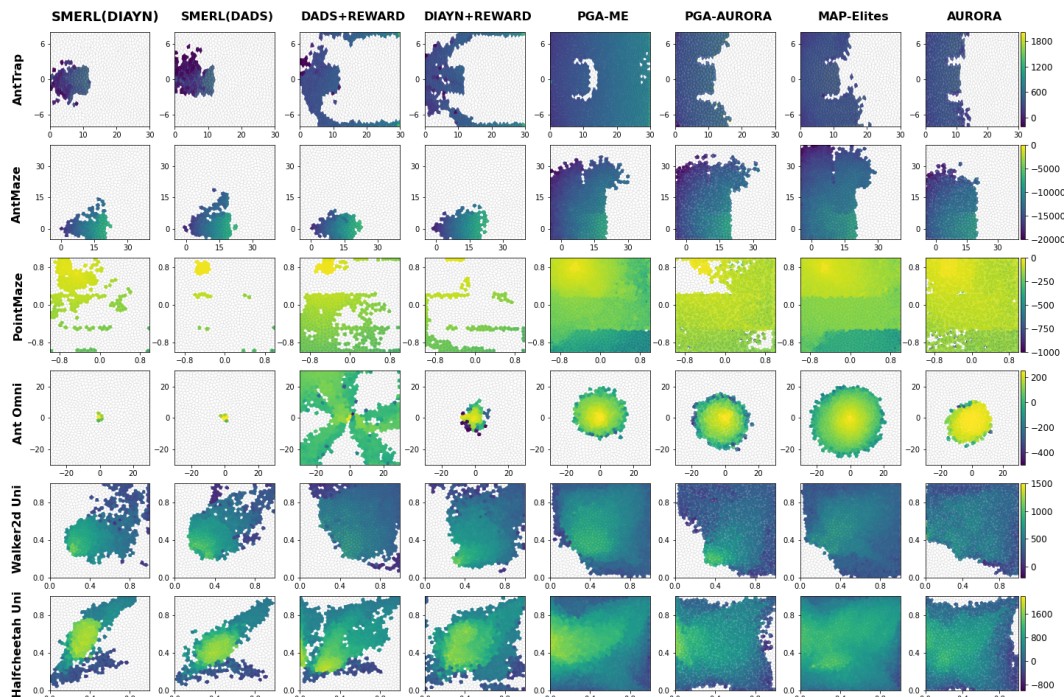

Figure 2: Visualizations of repertoires post training. Each repertoire divides the behavior descriptor space into 1024 cells. Cells are left blank when no policy with the corresponding behavior descriptor has been found and are otherwise colored as a function of the best fitness achieved across policies with a matching behavior descriptor. For ANT-TRAP, ANT-MAZE, POINT-MAZE, and ANT-OMNI, the behavior descriptor is the $(x, y)$ position reached at the end of an episode: these plots highlight explored areas of the 2D plane. For HALFCHEETAH-UNI and WALKER-UNI, the descriptor is the fraction of time the robots' feet were touching the ground during an episode (one dimension per foot).

and HALFCHEETAH-UNI. The second subset, referred to as OMNIDIRECTIONAL locomotion tasks and with the reward signal defined as the opposite of the energy consumption, includes ANT-OMNI.

**Mixing locomotion and exploration tasks.** Following Pierrot et al. (2022); Parker-Holder et al. (2020); Frans et al. (2018); Shi et al. (2020), we consider two additional hybrid tasks built upon ANT that exhibit both continuous control and exploration challenges, making them particularly difficult even for state-of-the-art evolutionary and RL algorithms. In ANT-TRAP, the ant has to run forward while initially facing a trap consisted of three walls that prevents it from moving forward along the x-axis direction indefinitely. In ANT-MAZE, the ant is initially placed in a maze and the goal is to reach a specified target position.

## 5.2 EXPERIMENTS DIRECTLY EVALUATING DIVERSITY

On the one hand, metrics quantifying the amount of diversity in a collection of policies are lacking in the MI RL literature. Success in inducing diversity is instead typically assessed indirectly through performance on adaptation or hierarchical learning experiments where either the tasks are modified or new ones are introduced (Eysenbach et al., 2019; Kumar et al., 2020; Sharma et al., 2019). On the other hand, diversity metrics were introduced very early in the QD literature and have been refined over time since QD algorithms are primarily assessed on this basis. Hence, we use metrics developed in the QD literature and extend them to MI RL approaches to be able to compare the methods on the same ground. Noting that almost all QD metrics crucially rely on the definition of a behavior descriptor space (to characterize solutions) and the availability of a repertoire of solutions indexed with respect to this space for all methods, we specify behavior descriptors for all tasks and extend the concept of repertoires to MI RL methods.

**Behavior descriptor spaces.** For POINT-MAZE, ANT-TRAP, ANT-MAZE, and ANT-OMNI, following several prior QD and RL works (Pierrot et al., 2022; Hansen et al., 2021), the behavior descriptor of

Table 1: Maximum fitness and QD score at the end of the training phase (median over 5 seeds). Additional statistics (mean, standard deviation, and interquartile mean) are included in the Appendix.

|  | ANT-TRAP | ANT-MAZE | POINT-MAZE | ANT-UNI | ANT-OMNI | WALKER-UNI | HALFCHEETAH-UNI |
|---|---|---|---|---|---|---|---|
| **Maximum fitness** | | | | | | | |
| SMERL(DIAYN) | 4.49 e2 | -7.88 e3 | -4.00 e1 | **1.71 e3** | **2.49 e2** | 1.19 e3 | 1.89 e3 |
| SMERL(DADS) | 4.48 e2 | **-7.62 e3** | -8.10 e1 | 1.61 e3 | **2.49 e2** | 1.17e3 | 1.92 e3 |
| DADS+REWARD | **1.28 e3** | -7.81 e3 | -8.10 e1 | 1.24 e3 | 2.38 e2 | 6.30 e2 | 1.89 e3 |
| DIAYN+REWARD | 9.42 e2 | -8.13 e3 | -4.50 e1 | 1.34 e3 | 2.29 e2 | 1.00 e3 | **2.13 e3** |
| PGA-MAP-ELITES | 9.77 e2 | -9.32 e3 | **-2.30 e1** | 1.45 e3 | **2.49 e2** | **1.22 e3** | 1.75 e3 |
| PGA-AURORA | 4.72 e2 | -9.27 e3 | -6.3 e1 | 1.44 e3 | **2.49 e2** | 1.05 e3 | 1.47 e3 |
| MAP-ELITES | 3.81 e2 | -1.01 e4 | -2.5 e1 | 4.25 e2 | **2.49 e2** | 6.87 e2 | 1.23 e3 |
| AURORA | 4.05 e2 | -1.10 e4 | -2.35 e1 | 5.05 e2 | **2.49 e2** | 5.71 e2 | 1.21 e3 |
| **QD score** | | | | | | | |
| SMERL(DIAYN) | 1.02 e5 | 7.84 e5 | 38.9 e4 | 6.20 e4 | 1.95 e3 | 3.01 e5 | 7.92 e5 |
| SMERL(DADS) | 9.88 e4 | 8.22 e5 | 3.57 e3 | 0.83 e5 | 4.15 e3 | 2.69 e5 | 7.51 e5 |
| DADS+REWARD | 2.42 e5 | 9.12 e5 | 0.42 e4 | 2.88 e5 | 3.05 e5 | 3.89 e5 | 8.31 e5 |
| DIAYN+REWARD | 2.27 e5 | 9.18 e5 | 4.04 e4 | 1.29 e5 | 4.00 e4 | 4.92 e5 | 1.68 e6 |
| PGA-MAP-ELITES | **7.89 e5** | 2.74 e6 | 3.92 e5 | **9.19 e5** | 1.58 e5 | **8.23 e5** | 2.98 e6 |
| PGA-AURORA | 4.40 e5 | 2.59 e6 | 2.88 e5 | 7.33 e5 | 1.69 e5 | 5.10 e5 | 2.39 e6 |
| MAP-ELITES | 4.53 e5 | **2.98 e6** | 4.22 e5 | 9.08 e5 | 3.04 e5 | 6.45 e5 | **3.05 e6** |
| AURORA | 3.54 e5 | 2.18 e6 | **4.58 e5** | 5.82 e5 | **4.00 e5** | 4.59 e5 | 2.45 e6 |

a policy is defined as the $(x, y)$ position of the robot's center of gravity at the end of an evaluation episode. For UNIDIRECTIONAL tasks, the $i$-th component of the behavior descriptor vector is defined as the proportion of time during which the $i$-th foot of the robot is in contact with the ground in an evaluation episode. This is a simple but effective way to capture the gait of a robot as demonstrated in Cully et al. (2015) where it is used to help robots adapt after having some of their legs damaged.

We stress that, in our experiments, MAP-ELITES and PGA-MAP-ELITES make use of the same behavior descriptors used to compute the QD metrics, which induces a bias in the analysis. This bias does not affect AURORA and PGA-AURORA as these methods build their own behavior descriptors in an unsupervised fashion. Nevertheless, for fairness, we embed the choice of the behavior descriptor space into MI RL methods whenever possible. Hence, in ANT-MAZE, ANT-TRAP, ANT-OMNI and POINT-MAZE, MI RL algorithms are provided with the prior that diversity w.r.t. the $(x, y)$ position of the robot is what needs to be optimized. Implementation details are deferred to the Appendix.

**Passive repertoires for RL methods.** The MI RL methods considered in this paper do not maintain a repertoire to store diverse solutions. For fairness, and since increasing the size of the discrete latent space to match the size of the QD repertoires makes these methods unstable (Eysenbach et al., 2019), we fill a repertoire on the side during training with all the intermediate policies found by these methods, similarly to what is done internally in QD methods. The repertoires of size 1024 are identically defined across methods using Centroidal Voronoi Tessellations (Vassiliades et al., 2017).

**QD Metrics.** Armed with behavior descriptor spaces for all environments and repertoires for all methods, we compute three metrics that are often used in the QD community to track the performance of a collection of solutions: (i) the **maximum fitness** of the repertoire, computed as the maximum fitness attained across the solutions stored in the repertoire, (ii) the **coverage** of the repertoire, computed as the number of cells (behavior niches) that have been filled, and (iii) the **QD score**, computed as the sum of fitnesses attained by the solutions stored in the repertoire.

**Results and analysis.** All methods are trained for exactly two hours on a Quadro RTX 4000 GPU for each environment described in Section 5.1. Each experiment is repeated with 5 seeds and the final repertoire is evaluated using the aforementioned QD metrics. The maximum fitness and QD score are reported in Table 1. Out of the 5 seeds, the best - in terms of QD score - final repertoires are depicted on Figure 2 for all methods and environments.

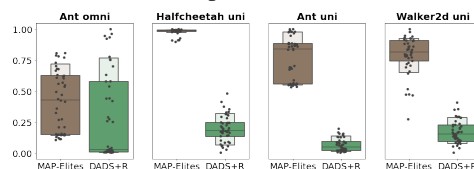

Figure 3: Distribution of QD scores obtained over a range of hyperparameters on UNIDI-RECTIONAL tasks and ANT-OMNI.

First, observe that, setting aside ANT-OMNI where DADS+REWARD has managed to largely explore the environment, QD methods outperform the MI RL methods for all environments and tasks if we base the comparison on QD scores and coverage only. This is a particularly desirable property in ANT-MAZE (resp. ANT-TRAP) because reaching a high coverage of the repertoire means that the

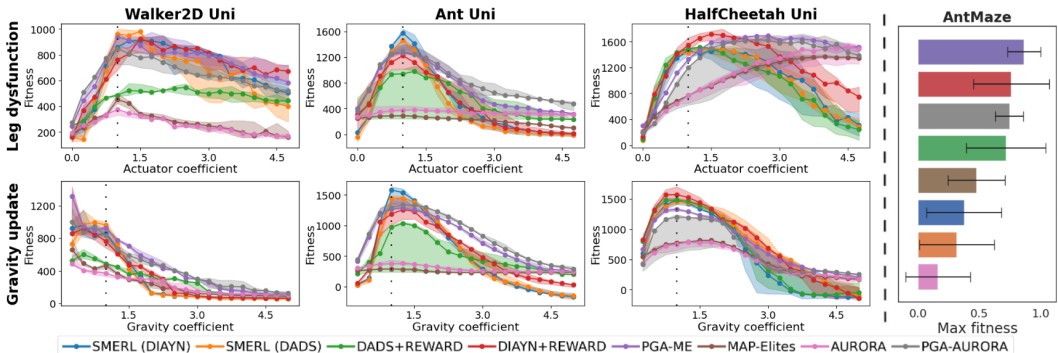

Figure 4: Maximum fitness of the methods when confronted to (left) a perturbed environment and (right) a modified task (median and interquartile range on 5 seeds, additional statistics are in the Appendix). We observe varying resilience levels across methods. In particular, PGA-MAP-ELITES and PGA-AURORA adapt better to extreme environment perturbations in UNIDIRECTIONAL environments.

method has been able to explore the maze (resp. get past the trap) effectively given that the behavior descriptor vector is defined as the final $(x, y)$ position of the robot.

Second, note that no single method outperforms all others in terms of maximum fitness. PGA-MAP-ELITES ranks first on POINT-MAZE and WALKER-UNI but is always outperformed by an MI RL method in the others. Additional observations are included in the Appendix.

While the excellent performance of DADS+REWARD on ANT-OMNI is an impressive achievement in itself, we run an additional set of experiments to investigate whether this may be the result of a lucky guess on the exact choice of hyperparameters. Specifically, we run the same experiments on ANT-OMNI and UNIDIRECTIONAL environments multiple times for DADS+REWARD and MAP-ELITES, sampling from a grid of reasonable hyperparameters values for both methods (exact details are deferred to the Appendix). The medians and quartiles of the QD-score distributions obtained are shown on Figure 3. We observe that the performance of DADS+REWARD is very sensitive to the exact choice of hyperparameters and that the median performance of MAP-ELITES is always significantly better than the median performance of DADS+REWARD on all tasks.

## 5.3 FEW-SHOT ADAPTATION EXPERIMENTS

While metrics borrowed from the QD literature provide a first basis for comparison, they suffer from an inherent bias in favor of QD methods as discussed in Section 5.2. Following prior work that analyze the resilience of RL methods to changes in the environment or task definition (Kumar et al., 2020), we design two series of few-shot adaptation experiments that do not favor any particular method a priori. Few-shot here refers to the fact that no re-training is allowed upon modification of the task or the environment, only evaluations of pre-computed policies are allowed. Throughout this section, MAP-ELITES and PGA-MAP-ELITES use the behavior descriptor spaces introduced in Section 5.2.

**Environment adaptation experiments.** In the first series of experiments, we modify a property of the UNIDIRECTIONAL environments after training, as illustrated on Figure 1. First, we vary the gravity coefficient in a neighborhood grid of size 20 around the default value (i.e. 1). We jointly refer to these experiments as the gravity-update setting. Second, we vary the actuators input-to-torque coefficients for a whole leg of the robot in a neighborhood grid of size 20 around the default value (i.e. 1). We jointly refer to these experiments as the leg-dysfunction setting.

**Task adaptation experiments.** In the second series of experiments, we modify the task of the ANT-MAZE environment by moving the target position after training, which directly modifies the reward signal. New target positions are sampled uniformly on the 2D plane delimited by the maze.

For both series of experiments, we train all methods on the nominal versions of the environments for exactly two hours and re-evaluate the final output of the training phase on the modified environment or task. Specifically, for each method, we enumerate all their learned skills, evaluate them one hundred times in the modified environment, and pick the one that performs best in terms of median fitness across evaluations. Each experiment is repeated with 5 different seeds and the maximal fitnesses obtained are reported on Figure 4 for the environment and task adaptation experiments.

**Results and analysis.** First, note that, on the environment adaptation experiments, PGA-MAP-ELITES and PGA-AURORA are performing on par with MI RL methods in the case of small perturbations, i.e. when the coefficient is perturbed by less than 50%. Methods that do not leverage policy gradients (i.e. MAP-ELITES and AURORA) are outperformed on all tasks in this regime.

Second, QD methods are more resilient than MI RL when confronted with significant perturbations. For instance, in the gravity-update setting, the relative performance loss is limited to 40% for PGA-MAP-ELITES when the gravity coefficient is divided by 4 compared to at least 60% for MI RL methods. As a testament of the resilience of QD methods, note that the best performing method is a QD one for 10 out of the 12 most extreme scenarios. Similar observations hold for ANT-MAZE where PGA-MAP-ELITES and PGA-AURORA perform as well as DIAYN+REWARD and DADS+REWARD.

Third, it is remarkable that PGA-AURORA and AURORA perform almost as well as their QD supervised counterparts in these experiments. This shows that unsupervised behavior descriptor learning is really effective for QD methods. Furthermore, observe that PGA-AURORA adapt with a similar loss in fitness than MI RL methods (recall that all of these methods have access to the exact same priors).

### 5.4 HIERARCHICAL LEARNING EXPERIMENTS

To further assess the skills discovered by the methods during training on a nominal environment, we introduce a hierarchical learning setting inspired from the RL literature (Eysenbach et al., 2019). In HALFCHEETAH-HURDLES, hurdles are spaced out along the x-axis direction in front of the robot's initial position in HALFCHEETAH, see Figure 1. The robot has to jump over them to move forward, the reward definition remaining unchanged. Just like in Eysenbach et al. (2019), we train a PPO meta-controller (Schulman et al., 2017) to choose a skill, which is to be unrolled in the environment for the next ten consecutive timesteps, among the ones learned by a given skill-discovery method during training on the nominal version of the environment, see Section 5.2. The meta-controller is trained for 200 million policy interactions, corresponding to 20 million actions of the meta-controller.

**Results and analysis.** The performance of the meta-controller is shown on Figure 5 for various choices of skill-discovery methods used to generate primitives. By visualizing the meta policies in the environment, we observe that PPO is able to learn a meta-controller that consistently manages to jump over multiple consecutive hurdles when it is trained with primitives derived from PGA-MAP-ELITES and DIAYN+REWARD, which achieve similar performance on this experiment. Surprisingly, all other skill-discovery methods fail to provide primitives that can be composed to make the robot jump over hurdles.

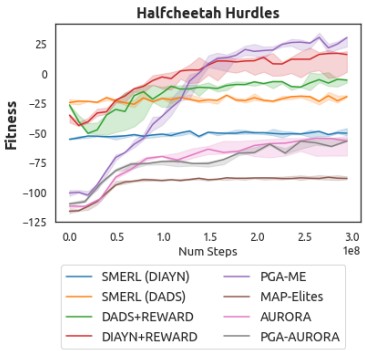

Figure 5: Fitness of the meta-controller as a function of environment steps for all types of skills primitives (median on 5 seeds).

## 6 DISCUSSION AND CONCLUSION

Our experiments show that QD methods are competitive with MI RL methods when the goal is to find a set of diverse and high-performing policies in continuous control and exploration environments. MI RL methods are task-agnostic and data-efficient but struggle to generate large collections of policies. Moreover, they are limited in the set of descriptors priors they can accommodate. QD methods are less data-efficient but can better leverage the vectorization capabilities of modern libraries such as JAX. Moreover, they are significantly less sensitive to the choice of hyperparameters than MI RL methods and their ability to evolve large sets of diverse solutions is a precious tool for hierarchical learning and adaptation applications when the task definition changes or when confronted with extreme environment perturbations. Our benchmarks however show that QD methods do not outperform MI RL methods and vice-versa, which opens the door to promising work directions. We believe that MI RL methods could make use of skill repertoires to improve their memory capacity thereby increasing the pool of available options for downstream applications. Conversely, QD methods leveraging gradients, such as PGA-MAP-ELITES, could benefit from the introduction of intrinsic rewards described in the MI RL literature to guide the search for diversity. Ultimately, we believe that combining the coverage capacity of a repertoire with the data-efficiency of gradient-based techniques is key to develop robust, diverse, and high-performing repertoires of solutions for challenging continuous control.

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

## A   IMPLEMENTATION DETAILS

In this section we detail the implementation choices that have been made to compare the algorithms. All our implementations are based on the JAX framework Bradbury et al. (2018) and are specifically designed to run on an hardware accelerator, especially with the accelerator-compatible environments of BRAX (Freeman et al., 2021).

Table 2: Identification of components shared across methods.

| | Algorithm | Genetic Mutation | Map-Elites-based | PG update | SAC-based | TD3-based |
|---|---|---|---|---|---|---|
| RL | SMERL(DIAYN) | X | X | ✓ | ✓ | X |
| | SMERL(DADS) | X | X | ✓ | ✓ | X |
| | DADS+REWARD | X | X | ✓ | ✓ | X |
| | DIAYN+REWARD | X | X | ✓ | ✓ | X |
| QD | PGA-MAP-ELITES | ✓ | ✓ | ✓ | X | ✓ |
| | PGA-AURORA | ✓ | ✓ | ✓ | X | ✓ |
| | MAP-ELITES | ✓ | ✓ | X | X | X |
| | AURORA | ✓ | ✓ | X | X | X |

| | Algorithm | Unsupervised diversity | SMERL sum. | DADS diversity | DIAYN diversity |
|---|---|---|---|---|---|
| RL | SMERL(DIAYN) | ✓ | ✓ | X | ✓ |
| | SMERL(DADS) | ✓ | ✓ | ✓ | X |
| | DADS+REWARD | ✓ | X | ✓ | X |
| | DIAYN+REWARD | ✓ | X | X | ✓ |
| QD | PGA-MAP-ELITES | X | X | X | X |
| | PGA-AURORA | ✓ | X | X | X |
| | MAP-ELITES | X | X | X | X |
| | AURORA | ✓ | X | X | X |

### A.1   ALGORITHMS UNDER STUDY

**MAP-Elites**   Our implementation of MAP-ELITES takes advantage of the just-in-time compilation and vectorization capacity of JAX, similarly to the implementation presented in QDAX (Lim et al., 2022). All environments are implemented in BRAX (itself constructed on JAX), which enables to simulate multiple copies of the same environment in parallel, on the same device. Hence, we can simultaneously unroll thousands of episodes and evaluate as many policies. In addition, we put focus on implementing the mutation and insertion steps of MAP-ELITES to be compatible as well with the just-in-time compilation paradigm of JAX. These design choices enable parallelization and lead to an overall speedup of several orders of magnitude compared to previous sequential implementations. Finally, we opt for the Iso+LineDD variation (Vassiliades & Mouret, 2018) for the updates of the population.

**PGA-MAP-Elites**   We provide here the first implementation of PGA-MAP-ELITES in JAX. This implementation benefits from the same vectorization strategies as MAP-ELITES which yields similar speedups. The major difference between these two algorithms lies in the policy variation step: rather than applying the Iso+LineDD mutation all the time, PGA-MAP-ELITES uses a mix of the latter variation and a policy gradient (PG) variation. As in Nilsson & Cully (2021), this PG variation consists in updating the selected policies to follow a TD3 critic trained with transitions from a shared replay buffer.

**DIAYN**   Our implementation closely follows the original implementation of Eysenbach et al. (2019). For the latent skill variable, we rely on a categorical prior with dictionary of size $|\mathcal{Z}|$ and we sample $z$ with probability $p(z) = \frac{1}{|\mathcal{Z}|}$. For a given transition $(s, z)$ from the replay buffer, the reward is computed online as follows:

$$r_t = \log(q(z|s)) - \log(p(z)) \qquad (1)$$

The discriminator $q(z|s)$ is parameterized as a neural network that outputs a softmax distribution and is trained as a classifier to predict the skill $z$ from the current state $s$. The discriminator is updated at the same frequency as the policy and the critic. However, one distinction with the original

implementation of DIAYN stems from the way we take advantage of the JAX's parallelization capacity: we sample multiple skills and simultaneously generate trajectories by unrolling the skill-conditioned policy on identical copies of the environment. This way, we collect data from different skills at the same time, and accelerate the training of the discriminator.

**DIAYN+REWARD and SMERL(DIAYN)** DIAYN+REWARD is a supervised RL method that consists in incorporating signal from the environment into the DIAYN algorithm. At each timestep, we sum the environment reward and the reward of DIAYN (multiplied by a scaling coefficient), see Equation 3. Thus, we expect to discover diverse and high-performing skills. SMERL(DIAYN) rather formulates the problem in a constrained manner: diversity should be maximized only when a policy is $\epsilon$-close to the return of an *optimal* policy, $\epsilon$ being a margin set as an hyperparameter. To implement this constraint, we maintain for each transition in the replay buffer the return of its associated episode. Only transitions from an $\epsilon$-optimal episode are updated to add the diversity reward.

**DADS** Our implementation of DADS builds on the same basis as the original implementation. The prior chosen for $z$ is the same as in DIAYN and the reward is defined as follows:

$$r_t = \log(q(s_{t+1}|s_t, z)) - \log(p(s)) \tag{2}$$

where $p(s)$ is obtained by marginalizing over the (discrete) distribution of $z$. Following the work from Sharma et al. (2019), we define the skill-dynamics model as a neural network which outputs a Gaussian Mixture distribution, with 4 experts and an identity covariance matrix. We also predict $\Delta s = s_{t+1} - s_t$ instead of $s_{t+1}$ and we exclude $s_t$ from the input, resulting in a skill-dynamics network of the form $q(\Delta s|z)$. We maintain the running average and the standard deviation of the target $\Delta s$ to normalize the output of the skill-dynamics network. Our implementation trains the skill-dynamics network offline, similarly to the training of DIAYN's discriminator, with parameters updated at the same frequency as the policy and the critic. Finally we take advantage of the batching capability of JAX along the same lines as for DIAYN.

**DADS+REWARD and SMERL(DADS)** The summation method of DIAYN+REWARD (resp. DIAYN+REWARD) can be easily extended with any MI method that provides a diversity reward. It naturally leads to the method DADS+REWARD (resp. SMERL(DADS)).

**AURORA** AURORA is QD algorithm introduced in Cully (2019). In this algorithm, the population is stored in an unstructured archive instead of a structured grid similarly to what is done in MAP-ELITES. The main difference of AURORA compared to usual QD methods is that the behavior descriptor is learned instead of being manually defined by the user. To do so, an auto-encoder learns to encode and to reconstruct trajectories collected in the environment. The latent space defines the behavior space. In order to determine the behavior descriptor of a policy, its trajectory in the environment is given as input to the encoder. The output of the encoder is the behavior descriptor of the policy. The auto-encoder is trained following a simple geometric schedule (with decreasing frequency). At each training step, the auto-encoder learns to reconstruct the trajectories of the policies that are stored in the archive. Then, the behavior descriptors are computed with the new weights of the encoder and the unstructured archive is updated accordingly. The genetic mechanism used to generate offspring in our study is the same as MAP-ELITES: we use the Iso+LineDD variation.

**PGA-AURORA** PGA-AURORA is a new algorithm that we introduce in this paper. It has components inspired by AURORA and PGA-MAP-ELITES. It uses the unsupervised behavior descriptor definition from AURORA, as well as its unstructured archive. But instead of using purely genetic mutations like AURORA, PGA-AURORA uses the policy-gradient mutation introduced in PGA-MAP-ELITES (Nilsson & Cully, 2021). This baseline is interesting because it has exactly the same prior knowledge as the MI RL methods considered in this benchmark: it does not rely on the definition of a behavior descriptor but a prior can be injected. In our experiments, the population of the unstructured archive is the same as for the other QD methods and the latent space size is set to 5. The auto-encoder uses LSTM units to encode and decode the given trajectories. The implementation of PGA-AURORA is also fully implemented in JAX.

## A.2 Passive repertoire: measuring diversity for latent-conditioned policies

Skill-discovery RL methods considered in this paper do not maintain a repertoire to store diverse solutions. Instead, policies are indexed by the possible values that the discrete latent variable can take for DIAYN, DADS, and SMERL. For these methods, a possible approach to introduce the same type of repertoire used by QD methods is to fill one in an ad-hoc fashion (i.e. post training) by enumerating all possible values for the latent variable and evaluating the policy conditioned on this value each time. However, this puts these methods at a clear disadvantage if the cardinality of the latent space is small compared to the size of the repertoire given that most QD metrics are increasing functions of the number of non-empty cells. We could increase the size of the discrete latent space to match the size of the repertoire but skill-discovery RL methods become unstable when the size of the latent space is large (Eysenbach et al., 2019). Instead, we choose to fill the repertoire during training with all the intermediate policies found by skill-discovery RL methods, in the same fashion as what is done internally in QD methods. Specifically, every 100,000 training step, we evaluate the fitness and behavior descriptors of all of the latent-conditioned policies and insert them in the repertoire whenever possible, following the same insertion rules as MAP-ELITES: if the cell corresponding to the behavior descriptor value is empty or if the solution stored previously in this cell has lower fitness. We stress that, in our experiments, the QD methods make use of the same behavior descriptors used to compute the QD metrics, which induces a bias in the analysis. To make the study less biased, we embed the choice of the behavior descriptor space into skill-discovery RL methods whenever possible. This is possible for OMNIDIRECTIONAL environments where we can choose a discriminator function for all skill-discovery methods that is conditioned on the behavior descriptor as the latter is just a function of the current state. This is however not possible for UNIDIRECTIONAL environments.

## A.3 Details on QD-score computation

As commonly done in the QD literature, we add an offset to the fitnesses when computing the QD score to guarantee that it is an increasing function of the coverage. The repertoires, which are initialized identically across methods, are of size 1024 and are constructed using Centroidal Voronoi Tessellations (Vassiliades et al., 2017).

## A.4 Environments and modifications

To define our environments, we use the BRAX physics simulator, which is provided in an open-source package fully implemented in JAX Freeman et al. (2021). BRAX builds environments from configurations that describe the elements to simulate. Specifically, the properties of the body parts of the agent, its joints, and its actuators, as well as the types of contacts that need to be taken into account during the simulations and a set of physic constants (including friction and gravity) need to be specified in the configuration.

Implementations for the base environments mentioned in section 5.1, (i.e. ANT, WALKER, and HALFCHEETAH) are provided in the BRAX package. The tasks derived from these environments (i.e. UNIDIRECTIONAL, ANT-OMNI, ANT-MAZE, ANT-TRAP) can be implemented with minor code changes. In UNIDIRECTIONAL, we pass along the feet/ground contact information from the simulator to the user to be able to compute the behavior descriptors. In ANT-OMNI, we do the same for the (x, y) position of the center of gravity of the robot. In ANT-TRAP, three walls are added in front of the ant to create the trap and the (x, y) position of the center of gravity is made available. Similarly, we have created a maze with simulated walls to generate the ANT-MAZE task.

In POINT-MAZE, actions are bounded 2-dimensional vectors corresponding to the $(x, y)$ increments that can be added to the current position. The observation received at each timestep is the current point's position on the 2D plane. The reward signal is defined as the negative euclidean distance between the current position and the target position.

In ANT-TRAP, the reward signal is defined as the x-axis velocity minus the energy consumption. In ANT-MAZE, the reward signal is defined as the negative euclidean distance between the ant's center of gravity and the target position.

For adaption experiments, we took inspiration from both the QD and the RL literature. The experiment consisting in introducing leg damage has been studied in Cully et al. (2015) and Kumar et al. (2020). Experiments that involve changing the gravity property have been used in Gaya et al. (2021). A major

difference in this work lies in the fact that we study a significantly larger range of perturbations in each setting, which enables us to get a sense of the adaptation capabilities of each method against both small and significant perturbations. The adaptation task for ANT-MAZE is also inspired from prior work in the QD literature (Pierrot et al., 2022).

In order to create variations of the environment in our adaptation tasks, we apply wrappers to modify a specific entry of the configuration. For instance, to change the gravity value, we (1) retrieve the configuration of the environment, (2) fetch the value of the gravity constant, (3) multiply it by a given input and (4) set the new value in the configuration. Similarly, we change the behavior of the actuator of a specific joint by updating the "actuator_strength" entry in the BRAX configuration. In order to create the leg-dysfunction adaptation task, we pick a multiplicative factor and multiply the "actuator_strength" of all the joints that link bodies of the concerned leg by this factor. A given action input will result in a different update of the joint angle in the resulting physical simulation compared to the original simulation. At the lower extreme of the range (i.e. 0), this makes the actuator completely ineffective. At the higher extreme of the range (i.e. 4.5), the joint swings much faster than in the nominal case for a given input command.

Below we detail several elements to change in the BRAX configurations in order to replicate our adaptation tasks. We also provide the code to replicate those.

- Leg dysfunction in WALKER-UNI: the leg impacted by the dysfunction is the left leg of the WALKER. The impacted joints in BRAX configuration are called "thigh_left_joint", "leg_left_joint" and "foot_left_joint".

- Leg dysfunction in ANT-UNI: the leg impacted by the dysfunction in the ant corresponds to the following joints in BRAX configuration: "$ Torso_Aux 4" and "Aux 4_$ Body 13".

- Leg dysfunction in HALFCHEETAH-UNI: the leg impacted by the dysfunction is the front leg of the HALFCHEETAH. The impacted joints in the BRAX configuration are called "fthigh", "fshin" and "ffoot".

- Gravity update in all the UNIDIRECTIONAL tasks: the gravity parameter of the environment is usually -9.8 by default. For each environment, we multiply it by a range of 20 values, to get values ranging from -2.45 (low gravity) to -490 (high gravity).

- Target position update in ANT-MAZE: we sampled 10 new positions uniformly inside the maze (i.e. in $[-3, 38]^2$). The values used in our final experiments are reported on Table 3.

For the hierarchical learning task, we have added the $(x, y)$-position of the robot to the observation because it is needed by the meta-controller to take relevant actions.

| Run number | x position | y position |
|---|---|---|
| Training | 35.0 | 0.0 |
| Adaptation 1 | 11.74 | 35.16 |
| Adaptation 2 | $-2.33$ | 14.86 |
| Adaptation 3 | 32.48 | 32.37 |
| Adaptation 4 | 29.52 | 17.46 |
| Adaptation 5 | 12.46 | 7.05 |
| Adaptation 6 | 17.36 | 4.45 |
| Adaptation 7 | 8.38 | 25.17 |
| Adaptation 8 | 24.02 | 34.36 |
| Adaptation 9 | 21.75 | 14.12 |
| Adaptation 10 | 19.51 | 14.58 |

Table 3: New target positions in ANT-MAZE adaptation task. During the training phase, the position is set to (35, 0) but varies during adaptation. The 10 different values used during our adaptation experiments are reported.

For the HALFCHEETAH-HURDLES, we add obstacles in front of the halfcheetah in the x-direction. To do so, we add capsule body to the Brax simulation configuration and include collisions between the

robot's body and the capsules. Those hurdles are placed every three meters, have a length (y-axis) of 1 meter, a height (z-axis) of $0.25$ meter and a radius of $0.25$ meter.

## B  HYPERPARAMETERS

In this section, we document the various hyperparameters that have been used in the study and the selection process that led us to pick these values.

| Hyperparameter | Value |
|---|---|
| Policy learning rate | 0.0003 |
| Critic learning rate | 0.0003 |
| Environment batch size | 200 |
| Batch size | 256 |
| Discount factor | 0.99 |
| Entropy coefficient | 0.1 |
| Policy hidden layers size | $[256, 256]$ |
| Critic hidden layers size | $[256, 256]$ |
| DIAYN | |
| Discriminator learning rate | 0.0003 |
| Discriminator hidden layers size | $[256, 256]$ |
| Number of discrete skills | 5 |
| DADS | |
| Skill-dynamics learning rate | 0.0003 |
| Skill-dynamics hidden layers size | $[256, 256]$ |
| Number of discrete skills | 5 |

Table 4: Hyperparameters for DIAYN and DADS. Most parameters are inspired from the original papers that introduced the methods.

**Skill-discovery MI RL methods.**  Throughout our evaluation, we use the same hyperparameter values for shared parameters across all methods and environments. Table 4 details the choice of hyperparameters for DIAYN and DADS, which both use SAC internally. This set of hyperparameters values, derived from Osa et al. (2022) and Kumar et al. (2020), yields good performance across all environments for both methods. However, some environment-dependent parameters require more care. In particular, the global reward multiplier of the environment is set to $1.0$ for all environments except for HALFCHEETAH-UNI (resp. ANT-UNI), where it is set to $5.0$ (resp. $10.0$), following observations reported in BRAX (Freeman et al., 2021). One crucial parameter choice for DIAYN+REWARD and DADS+REWARD (and by extension SMERL(DIAYN) and SMERL(DADS)) is the **diversity reward scale**, $\beta$, that affects the diversity term in the weighted sum:

$$r_t = r_t^{\text{environment}} + \beta r_t^{\text{diversity}} \tag{3}$$

We set $\beta = 2.0$ as it has proved to favor the emergence of diverse and high-performing skills over all environments in our experiments, setting aside ANT-MAZE, where we set $\beta = 3.0$ to adjust to the larger scale of the environment's rewards, and ANT-OMNI, where we set $\beta = 4.0$ to compensate for the fact that the environment's reward drives the agent to learn static strategies by penalizing the use of the motors. The same values of $\beta$ were used for SMERL. To select the *optimal* SMERL return, similarly to Kumar et al. (2020), we evaluate a SAC agent on 5 seeds and select the median of the best-return policy as the target. The margin-to-optimal return is set as $10\%$ of the target return. For all DIAYN and DADS variants, we select a discrete skill distribution of size $|\mathcal{Z}| = 5$, following choice made in SMERL (Kumar et al., 2020). It is often hard to learn bigger numbers of distinct skills (Eysenbach et al., 2019).

Finally, the value for environment batch size was set to 200. To select it, we tried a range of values similarly to what was done in BRAX paper (Freeman et al., 2021). In the end, 200 was the highest value that sped up the training as much as possible while not perturbing the learning dynamics.

| Hyper-parameter | Value |
|---|---|
| Environment batch size | 100 |
| Policy learning rate | 0.001 |
| Critic learning rate | 0.0003 |
| Policy hidden layers size | $[256, 256]$ |
| Critic hidden layers size | $[256, 256]$ |
| Policy noise | 0.2 |
| Noise clip | 0.5 |
| Discount | 0.99 |
| Reward scaling | 1.0 |
| Policy gradient proportion | 50% |
| Critic training steps | 300 |
| Policy training steps | 100 |
| Iso sigma | 0.005 |
| Line sigma | 0.05 |

Table 6: Hyperparameters for PGA-MAP-ELITES. Most parameters values are standard in the literature on MAP-ELITES and TD3 for the tasks considered in this paper. For fairness, we used the same architecture for all methods under study whenever possible. The number of policy gradient steps is increased compared to the original implementation as this has proved to improve performance.

**QD methods** For PGA-MAP-ELITES, we took the hyperparameter values used in the original implementation (Nilsson & Cully, 2021), see Table 6, setting aside the number of PG steps applied to a policy when it is updated by a policy gradient variation. Instead of the value 10 used in the original paper, we opted for a value of 100 which yielded better performances in practice in our experiments.

| Hyperparameter | Value |
|---|---|
| Environment batch size | 1000 |
| Policy hidden layers size | $[256, 256]$ |
| Iso sigma | 0.005 |
| Line sigma | 0.05 |

Table 5: Hyperparameters for MAP-ELITES.

Note that we pick a different value for the environment batch size for MAP-ELITES and PGA-MAP-ELITES. This hyperparameter corresponds to the number of policies that are evolved and evaluated in parallel at each step of the algorithm. For MAP-ELITES, empirical evidence from a previous study, namely Lim et al. (2022), suggests that a large environment batch size dramatically speeds up the training phase without impacting the performance. Hence we chose to batch 1000 environments in parallel in order to leverage as much as possible the vectorization capabilities of JAX.

However, there is no such empirical evidence for PGA-MAP-ELITES. Specifically, since some elements are shared across the policies trained in parallel (e.g. the replay buffer of the underlying TD3), scaling up the environment batch size may negatively impact the training. As a matter of fact, multiplying by 10 the environment batch size for PGA-MAP-ELITES will not result in an 10-fold increase of the number of environment steps collected in the same amount time (which would be the case for MAP-ELITES). For this reason, we decided to keep the exact same value of environment batch size as in the original PGA-MAP-ELITES study (i.e. 100).

| Hyperparameter | Value |
|---|---|
| Latent space size | 5 |
| Archive initial l value | 0.2 |

Table 7: AURORA-specific hyperparameters for AURORA. The other hyperparameters are similar to MAP-ELITES.

## C   EVOLUTION OF QD METRICS DURING TRAINING

In this section, we report some additional results which shed light on the evolution of performance during training w.r.t. the QD metrics described in Section 5. In the main body of this paper, the final performances (QD score and maximum fitness) achieved by all methods after two hours of training are reported in Table 1 while visualizations of the best final grids obtained for each method are also available in Figure 2. In this section, we plot the evolution of the QD metrics during training for all methods in Figure 6. Additionally, Table 8 reports the number of environment steps carried out during training, averaged across the 8 evaluated environments, for all methods.

In our study, we chose to express the training budget in terms of time instead of environment steps. As hardware accelerators are getting more efficient, we believe that the intrinsic capacity of an algorithm to make the most of the available hardware is an important property to assess. Furthermore, with the development of highly accurate simulators, many real-world applications can be simulated, which enables to break down the training phase in two steps: (1) training in simulation (where the number of environment interactions is irrelevant, only the time budget is), and (2) sim-to-real training; rather than a single training phase in the real world.

| Algorithm | Env. steps |
|---|---|
| SMERL(DIAYN) | $1.0 \times 10^7$ |
| DIAYN+REWARD | $1.0 \times 10^7$ |
| SMERL(DADS) | $8.5 \times 10^6$ |
| DADS+REWARD | $8.5 \times 10^6$ |
| PGA-MAP-ELITES | $1.8 \times 10^8$ |
| MAP-ELITES | $1.6 \times 10^9$ |
| AURORA | $1.2 \times 10^9$ |
| PGA-AURORA | $1.1 \times 10^8$ |

Table 8: Number of training steps carried out (on average) during training by the various methods under study.

Since sample-efficiency is critical for (2), we also provide plots showing the evolution of the QD metrics w.r.t. the number of environment interactions during training. We plot the evolution of QD metrics for all QD (resp. skill-discovery) methods as a function of the number of environment steps, with a maximum horizon of $1.5 \times 10^8$ (resp. $10^7$) steps, on Figure 7 (resp. Figure 8). Figure 7 highlights the gain in sample efficiency yielded by the PG variation in PGA-MAP-ELITES compared to MAP-ELITES (although MAP-ELITES performs well on a time scale, see Figure 6).

We also provide the standard deviation of the maximum fitness and QD-score reported in Table 1 in Table 9.

Table 9: Maximum fitness and QD score at the end of the training phase (median over 5 seeds). The standard deviation is reported.

| | ANT-TRAP | ANT-MAZE | POINT-MAZE | ANT-UNI | ANT-OMNI | WALKER-UNI | HALFCHEETAH-UNI |
|---|---|---|---|---|---|---|---|
| | | | **Maximum fitness** | | | | |
| SMERL(DIAYN) | 4.49 (± 0.01)e2 | -7.88 (± 0.41)e3 | -4.00 (±2.57)e1 | **1.71 (±0.09)e3** | 2.49 (±0.00)e2 | 1.19 (±0.06)e3 | 1.89 (±0.14)e3 |
| SMERL(DADS) | 4.48 (±0.02)e2 | **-7.62 (±0.43)e3** | -8.10 (±3.19)e1 | 1.61 (±0.18)e3 | **2.49 (±0.00)e2** | 1.17(±0.14)e3 | 1.92 (±0.21)e3 |
| DADS+REWARD | **1.28 (±0.41)e3** | -7.81 (±0.49)e3 | -8.10 (±3.20)e1 | 1.24 (±0.52)e3 | 2.38 (±0.11)e2 | 6.30 (±0.48)e2 | 1.89 (±0.03)e3 |
| DIAYN+REWARD | 9.42 (±3.43)e2 | -8.13 (±0.51)e3 | -4.50 (±2.53)e1 | 1.34 (±0.21)e3 | 2.29 (±0.04)e2 | 1.00 (±0.13)e3 | **2.13 (±0.27)e3** |
| PGA-MAP-ELITES | 9.77 (±2.22)e2 | -9.32 (±0.43)e3 | **-2.30 (±0.03)e1** | 1.45 (±0.06)e3 | 2.49 (±0.00)e2 | **1.22 (±0.14)e3** | 1.75 (±0.04)e3 |
| PGA-AURORA | 4.72 (±0.31)e2 | -9.27 (±0.06)e3 | -6.3 (±2.3)e1 | 1.44 (±0.19)e3 | 2.49 (±0.00)e2 | 1.47 (±0.29)e3 | 1.47 (±0.06)e3 |
| MAP-ELITES | 3.81 (±0.11)e2 | -10.1 (±0.05)e4 | -2.5 (±0.05)e1 | 4.25 (±0.26)e2 | 2.49 (±0.00)e2 | 6.87 (±0.70)e2 | 1.23 (±0.06)e3 |
| AURORA | 4.05 (±0.52)e2 | -1.10 (±0.09)e4 | -2.35 (±0.12)e1 | 5.05 (±0.26)e2 | 2.49 (±0.00)e2 | 5.71 (±1.34)e2 | 1.21 (±0.08)e3 |
| | | | **QD score** | | | | |
| SMERL(DIAYN) | 1.02 (± 0.17)e5 | 7.84 (± 0.44)e5 | 38.9 (±13.8)e4 | 6.20 (± 1.38)e4 | 1.95 (±1.70)e3 | 3.01 (±0.20)e5 | 7.92 (±2.23)e5 |
| SMERL(DADS) | 9.88 (± 0.56)e4 | 8.22 (± 0.07)e5 | 3.57 (± 6.84)e3 | 0.83 (± 1.17)e5 | 4.15 (±1.11)e3 | 2.69 (±0.38)e5 | 7.51 (±1.60)e5 |
| DADS+REWARD | 2.42 (± 0.80)e5 | 9.12 (± 0.76)e5 | 0.42 (± 7.56)e4 | 2.88 (± 1.07)e5 | 3.05 (±1.22)e5 | 3.89 (±0.41)e5 | 8.31 (±3.91)e5 |
| DIAYN+REWARD | 2.27 (± 0.62)e5 | 9.18 (± 0.16)e5 | 4.04 (± 1.60)e4 | 1.29 (± 0.39)e5 | 4.00 (±0.39)e4 | 4.92 (±0.35)e5 | 1.68 (±0.18)e6 |
| PGA-MAP-ELITES | **7.89 (±2.87)e5** | 2.74 (± 0.50)e6 | 3.92 (± 0.03)e5 | **9.19(± 0.09)e5** | 1.58 (±0.47)e5 | **8.23 (±0.42)e5** | 2.98 (±0.52)e6 |
| PGA-AURORA | 4.40 (±0.64)e5 | 2.59 (±0.15)e6 | 2.88 (±0.80)e5 | 7.33 (±0.19)e5 | 1.69 (±0.42)e5 | 5.10 (±0.64)e5 | 2.39 (±0.06)e6 |
| MAP-ELITES | 4.53 (± 0.54)e5 | **2.98 (±0.68)e6** | 4.22 (± 0.01)e5 | 9.08 (±0.20)e5 | 3.04 (±0.74)e5 | 6.45 (±0.33)e5 | **3.05 (±0.00)e6** |
| AURORA | 3.54 (±0.11)e5 | 2.18 (±0.59)e6 | **4.58 (±0.72)e5** | 5.82 (±0.79)e5 | **4.00 (±1.33)e5** | 4.59 (±1.16)e5 | **2.45 (±0.04)e6** |

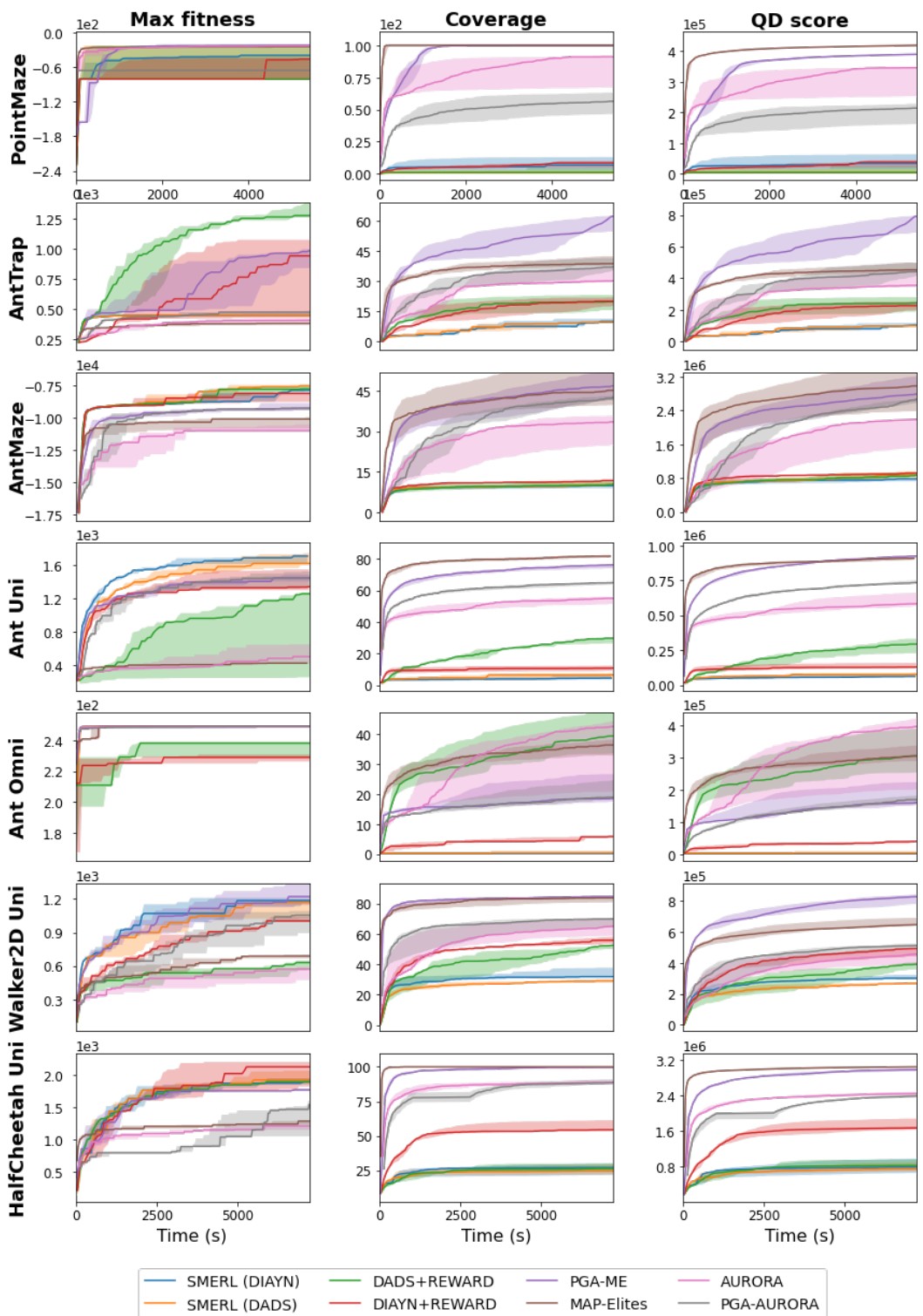

Figure 6: Evolution of the maximum fitness, coverage and QD score during the training phase (2 hours), for all methods under study.

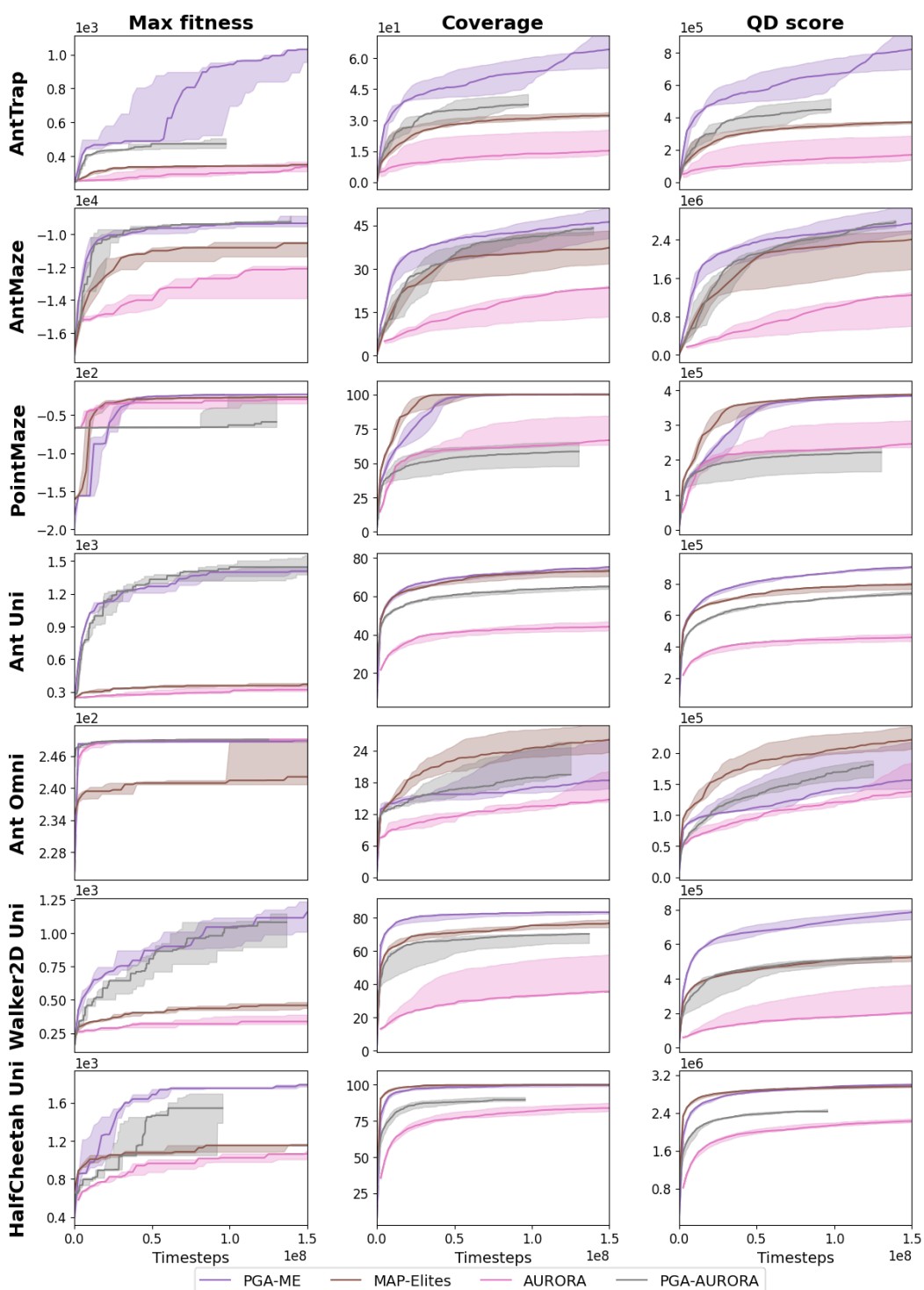

Figure 7: Evolution of the maximum fitness, coverage and QD score along environment interactions, during the training phase. Comparing QD algorithms MAP-ELITES, AURORA, PGA-MAP-ELITES and PGA-AURORA on $1.5 \times 10^8$ timesteps.

## D ADDITIONAL VISUALIZATIONS OF THE TRAINING PHASE

In this section, we provide additional visualizations of the training phase. Figure 11 display nine intermediate repertoires obtained by MAP-ELITES, PGA-MAP-ELITES and DADS+REWARD at various

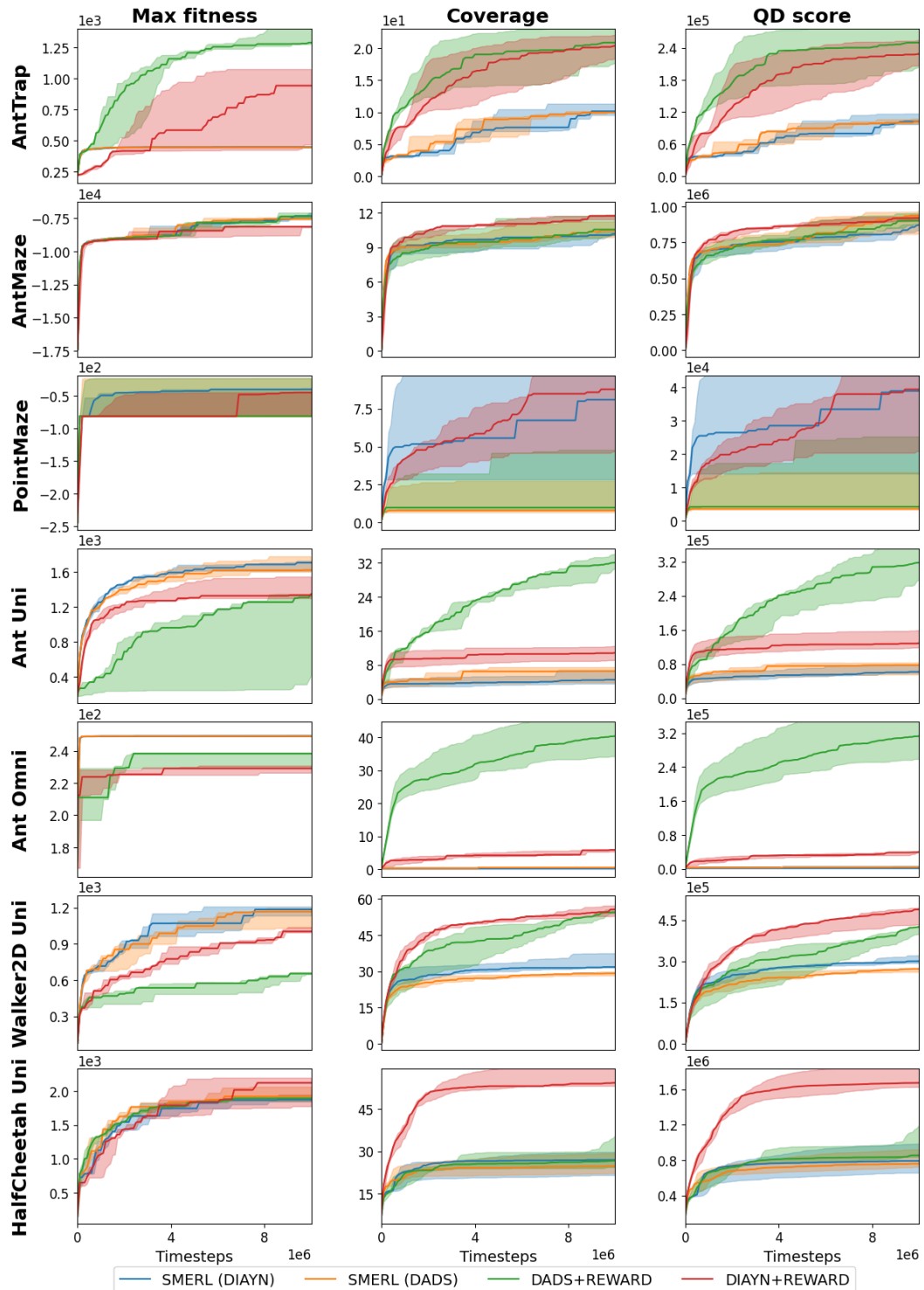

Figure 8: Evolution of the maximum fitness, coverage and QD score along environment interactions, during the training phase. Comparing skill-discovery algorithms DIAYN+REWARD, SMERL(DIAYN), DADS+REWARD and SMERL(DADS) on $10^7$ timesteps.

points during training, evenly distributed over the first hour of training. These visualizations shed some light on the mechanisms through which these repertoires are filled during training.

For instance, we observe that the use of a policy gradient in PGA-MAP-ELITES focuses the search toward solutions that move alongside the x-axis direction (recall that only x-axis movements are rewarded in the locomotion environments) whereas the first checkpoint for MAP-ELITES shows solutions that can go everywhere alongside the y-axis even though no solution has been able to hit the trap yet. MAP-ELITES struggles to further fill the repertoire (after 20M steps, the repertoire has almost reached its final state), whereas PGA-MAP-ELITES continuously improves the repertoire coverage to finally reach (nearly) 100% after two hours of training, see Figure 2. Finally, note that DADS+REWARD rapidly reach high-returns behaviors and focus on improving them through the training. However, it neglects intermediate policies that are sub-optimal but diversity-carrying.

We provide further comparison in the way the repertoire are filled by the different methods. Figure 12a highlights the capability of MAP-ELITES to fill the whole descriptor space, but it struggles getting excellent policies whereas SMERL(DIAYN) lacks coverage and can only evolve a small population of elite policies. On ANT-OMNI, the reward signal corresponds to the robot control cost, which hence gives incentives for minimal movement.

Nevertheless, we can see on Figure 12b how PGA-MAP-ELITES and DADS+REWARD eventually explore the environment. PGA-MAP-ELITES uses the divergent search of genetic mutations to gradually find exploratory behaviors and although it fails to go far from the origin, it is able to build a dense set of behaviors. On the other hand, DADS+REWARD uses the diversity rewards to discover far region of the behavior space through directed skills. Both methods have efficient strategies to keep exploring even if the environment rewards discourage them to do so.

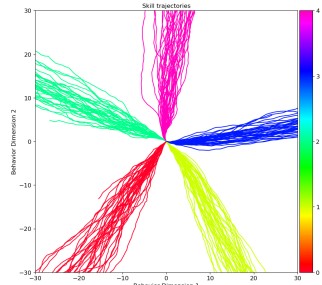

Figure 9: Visualization of final skills' trajectories (40 random seeds per skill) learned by DADS+REWARD on ANT-OMNI.

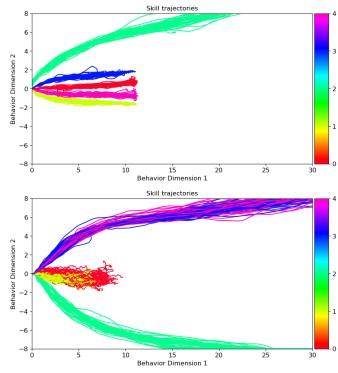

Figure 10: Visualizations of the final skills' trajectories (40 random seeds per skill) learned by DIAYN+REWARD (left) and DADS+REWARD (right) on ANT-TRAP.

While we have extended the concept of repertoires to skill-discovery methods in Section 5.2 to be able to compare all methods with the same metrics, an alternate way to assess the diversity achieved by the skill-discovery methods is to directly visualize the trajectories of the learned skills. On Figure 10, we show the trajectories of the 5 skills learned by DIAYN+REWARD and DADS+REWARD at the end of the training phase. On this particular seed, note that DIAYN+REWARD has been able to learn 5 skills that are significantly distinct from one another, with one of them able to sidestep the trap and run far along the x-axis direction. Conversely, DADS+REWARD could hardly learn 5 distinct skills but managed to find two significantly distinct skills that go around the trap on opposite sides. Nevertheless, Figure 9 shows how DADS+REWARD is able to learn skills going in 5 perfectly distinct directions in ANT-OMNI.

# E  ADDITIONAL OBSERVATIONS FOR THE TRAINING PHASE

It is worth mentioning that all skill-discovery RL methods achieve higher fitnesses than the QD methods on ANT-MAZE even though they fail to fully explore the behavior descriptor space, as seen on Figure 2. Amusingly, watching the policies trained by these methods interact with the simulator reveals that they all have found a way to jump above a wall of the maze, thereby directly reaching the target position without having to solve the maze. Although the task was not designed to be solved this way, it is interesting to see that some algorithms were able to find an original way to solve the

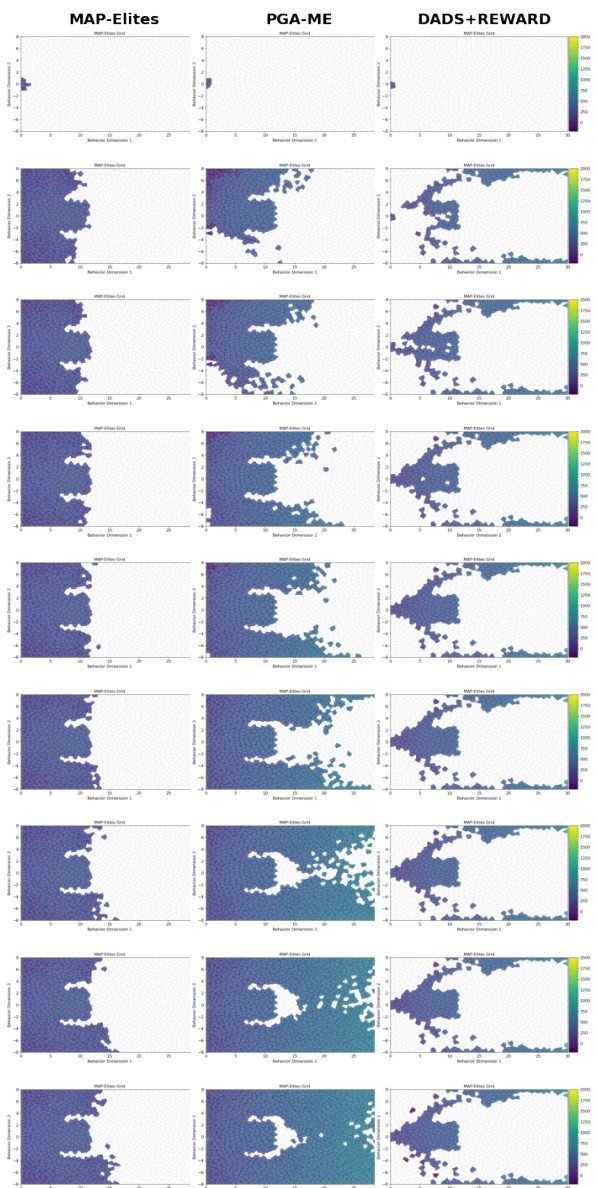

Figure 11: **Intermediate repertoires of MAP-Elites, PGA-MAP-Elites and DADS+REWARD on AntTrap.** Visualizations were generated at regular intervals from the start of training (top) to halfway through the training phase (i.e. at the 1h mark) (bottom).

task. Running a pure optimization RL algorithm with no mechanism for diversity does not find this way to jump above the maze.

It is interesting to see that Skill Discovery RL methods get best performance on 4 tasks collectively, but only get one best fitness each. No method has a consistent best performance on the tasks.

For the UNIDIRECTIONAL tasks, we observe that the skill-discovery methods have been able to illuminate an important portion of the space even though they are not explicitly looking for diversity w.r.t. the behavior descriptors. However, note that they have not been able to find solutions at the edges of the repertoire, whose cells can only be occupied by solutions consisting in walking on only one feet or in an asymmetric manner which we expect to be more robust to perturbations.

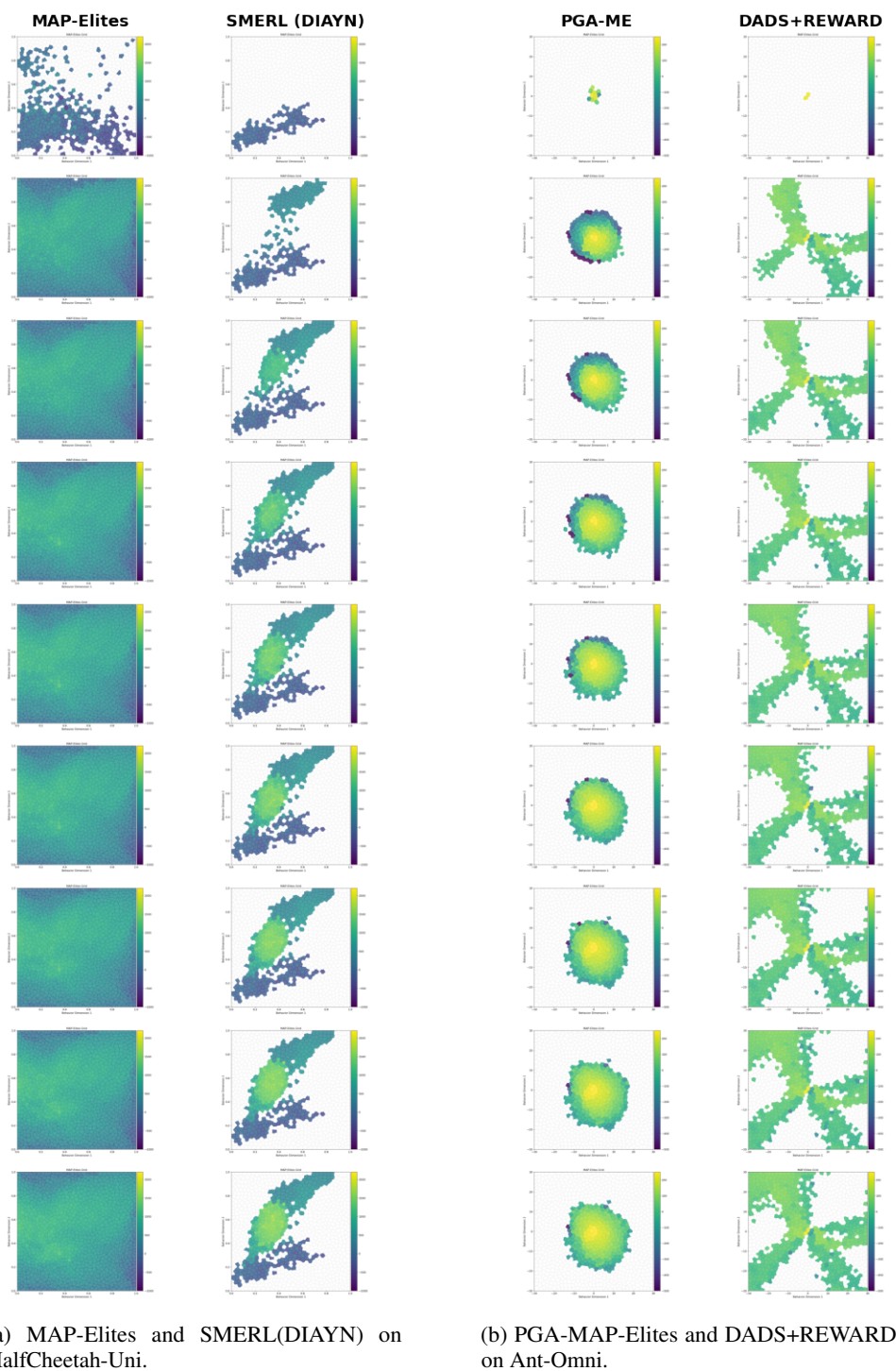

(a) MAP-Elites and SMERL(DIAYN) on HalfCheetah-Uni.

(b) PGA-MAP-Elites and DADS+REWARD on Ant-Omni.

Figure 12: Intermediate repertoires. Visualizations were generated at regular intervals from the start of training (top) to halfway through the training phase (i.e. at the 1h mark) (bottom).

Lastly, note that MAP-ELITES outperforms all other methods w.r.t. the QD score for 3 out of the 7 tasks while achieving similar coverage than PGA-MAP-ELITES, which means that it has been able to find high-performing solutions. This is a surprising fact in itself because this method does not leverage back-propagation through the neural policies so it is expected to be less efficient than the

other methods. This is a testament to the efficiency of the vectorized JAX implementation that enables MAP-ELITES to iterate over large populations of policies very quickly.

## F  ADDITIONAL RESULTS FOR THE ADAPTATION EXPERIMENTS

In this section, we provide additional results for the environment adaptation experiments described in Section 5.3. Specifically, we plot the relative change in fitness, referred to as *fitness gain*, compared to the nominal scenario (where the environment is unchanged) on Figure 13. This is to be compared with the raw fitness reported on Figure 4 in the main body of the paper. Observe on Figure 13 that the fitness gain is always 0 when the gravity (resp. actuator) coefficient is 1, this is by definition because the environment is unmodified in this case. We expect the fitness gain to be negative in most cases, because agents should perform worse on environments they were never trained on. Nevertheless, we can see that, in some cases (small gravity values or highly sensitive actuators), some methods find policies that can make the best of these changes and actually perform better than on the environment they were trained on.

Figure 13 also provides insights into the sensitivity of the various methods to environmental changes. MAP-ELITES, while lagging behind in terms of raw performance compared to other methods, is particularly resilient in the face of environment perturbations. For instance, for the ANT-UNI task, the performance of MAP-ELITES drops by at most 20% on the full range of gravity values used for evaluation, whereas the fitnesses of most methods drop by at least 70% in the most extreme conditions on both ends of the spectrum.

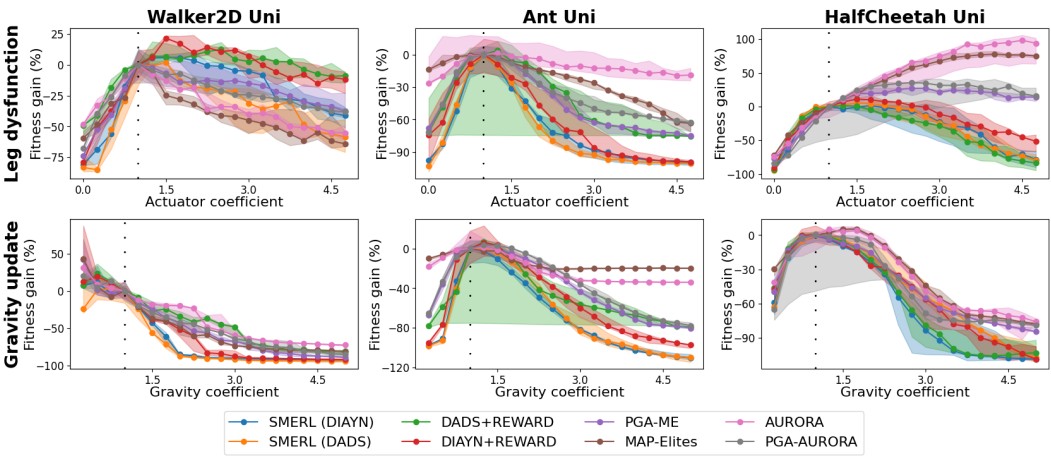

Figure 13: Relative change in fitness, referred to as *fitness gain*, on the adaptation tasks. Comparing the algorithms' performances on modified environments to their own performances on the nominal environments offers another perspective on the resilience capabilities of the methods under study.

## G  COMPARING SUPERVISED SKILL-DISCOVERY METHODS

This paper uses four methods inspired from the skill-discovery literature: DIAYN+REWARD and DADS+REWARD sum their diversity terms with the extrinsic reward from the environment, whereas SMERL(DIAYN) and SMERL(DADS) optimize the diversity reward only if the skill is already close to be optimal w.r.t. the extrinsic reward. We can draw three conclusions about the relative merits of each approach from the results obtained in this paper.

First, during the training phase, we can see, on Table 1 as well as on Figure 6, that SMERL methods outperform "sum"-based methods on most environments w.r.t. fitness, with the notable exception of ANT-TRAP where DADS+REWARD outperforms any other method by a significant margin. On the contrary, we can see that "sum"-based methods achieved higher QD scores than the SMERL methods on this benchmark. These results seem coherent with the fact that SMERL prioritizes optimizing for quality over optimizing for diversity. Nevertheless, this choice also impacts the

exploration capabilities of SMERL methods which, in turn, affects their fitness performances on some environments that require a good amount of exploration to be solved, such as ANT-TRAP. SMERL methods fail to find a way around the trap in ANT-TRAP whereas the "sum"-based methods eventually find one. Even if this improved exploration capability does not systematically lead to improved fitness in all environments, it helps "sum"-based methods attain higher QD scores.

Second, in the adaptation experiments, SMERL methods outperform "sum"-based methods as long as the perturbations remain small but do not necessarily perform better in extreme scenarios. On ANT-UNI for instance, SMERL(DIAYN) outperforms DIAYN+REWARD by a significant margin (more than 20% of maximum fitness, see Figure 4) when the actuator coefficient is perturbed by less than 25%, but for perturbations greater than 60%, DIAYN+REWARD eventually outperform (for smaller values) or draw (for bigger values) SMERL(DIAYN). We can observe similar tendencies on the gravity update experiment.

Third, "sum"-based methods seem to suffer from higher variances than SMERL methods. The performance of DADS+REWARD on ANT-UNI (see Figure 4) is an example of this tendency on the adaptation tasks. We also observed a significantly higher variance of DADS+REWARD and DIAYN+REWARD on several environments during the training phase, for all three metrics reported on Figure 6.

Although the benchmark carried out in this paper confirms the benefits of SMERL (Kumar et al., 2020) for adaptation tasks, it also shows that there is room for further improvements for these methods, particularly in settings where the task or the environment is significantly perturbed after training.

## H  HYPERPARAMETERS SENSITIVITY

In this section, we detail the hyperparameter-sensitivity experiment presented in Section 5.2 whose results are shown on Figure 3. Recall that this experiment was motivated by the fact that DADS+REWARD achieves impressive results on the ANT-OMNI task, covering a significant portion of the behavioral descriptor space, and outperforms all QD methods - let alone other skill-discovery methods, see Figure 2. Although it is not entirely surprising that DADS-related methods perform better than DIAYN-inspired methods, we were surprised to see such a discrepancy in the results and we suspected that the results would be rather sensitive to the exact choice of hyperparameters. For this reason, as described in Section 5.2, we decided to run a study on the hyperparameter sensitivy of DADS+REWARD on ANT-OMNI, comparing it to MAP-ELITES, the second most performing method in terms of QD score. To do so, we selected, for both methods, hyperparameters that impact the updates of the policies and we ran a grid search on them to compare the distributions of QD scores. For DADS+REWARD, we selected the hyperparameters that impact the reward signal, as they are key to trade off diversity for fitness, and used three different values for each hyperparameter, within an acceptable range, based on the literature. We picked the diversity reward scaling value (referred to as $\beta$ in Section B) in the set $\{0.1, 1, 10\}$. We picked the entropy temperature coefficient $\alpha$ in the set $\{0.1, 0.5, 1\}$, as it directly impacts the randomness of skills. For MAP-ELITES, the distribution of evolved policies is highly dependant on the exact definition of the genetic variations used as subroutines. Hence, we decided to run a grid search on the iso (resp. line) sigma values of the Iso+LineDD variation (Vassiliades & Mouret, 2018) used in MAP-ELITES. Iso sigma parameter were selected in the set $\{0.001, 0.01, 0.1\}$ and line sigma were selected in the set $\{0.01, 0.1, 1.0\}$. For each algorithm, this yields 9 different possibilities. Each possibility is evaluated on 5 different seeds. The results are reported on Figure 3.

In order to strengthen this study, we decided to run the same experiments on the three UNIDIRECTIONAL tasks as well: ANT-UNI, WALKER-UNI and HALFCHEETAH-UNI. We present the results on all the four environment on Figure 14.

On HALFCHEETAH-UNI, we can see that MAP-ELITES is being very consistent over the set of hyperparameters, with a high median and a very low variance, where DADS+REWARD has a low median and a rather high variance. On the WALKER-UNI, both methods have a similar variance over the hyperparameters but MAP-ELITES is consistently performing better. On ANT-UNI, MAP-ELITES has a significantly higher variance but it's worse case still outperforms DADS+REWARD's best QD score on the task. Those results shows that MAP-ELITES is easy to use off-the-shelf as it tends to have a smaller variance and as its worse case performance are usually decent. This makes this family algorithm very convenient for industrial use.

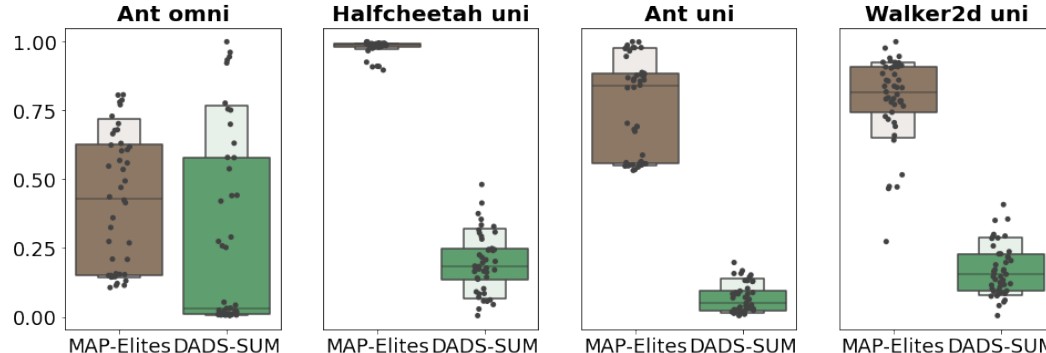

Figure 14: QD scores obtained when averaging over a grid of hyperparameters on ANT-OMNI, HALFCHEETAH-UNI, ANT-UNI and WALKER-UNI. QD scores are normalized with the maximal values obtained on each task. Each point represent one set of hyper-parameters. Main axes represent the 0.25 and 0.75 quantiles, the horizontal bar represents the median and the light color boxes represent the 0.125 and 0.875 quantiles.

## I    UNSUPERVISED BEHAVIOR DISCOVERY IN QUALITY DIVERSITY METHODS

The two Quality Diversity methods presented in the paper, MAP-ELITES and PGA-MAP-ELITES, are using a predefined behavior descriptor, which suppose some prior knowledge about the task. Skill-discovery RL methods often use a prior too (for instance, the xy position of the center of gravity, which is what we did in this study as well), they can be applied without it if needed.

Nevertheless, recent works from the QD community introduce methods to learn the behavior descriptor in an unsupervised manner (Cully, 2019; Grillotti & Cully, 2022a). One of these methods, called AURORA, stores the trajectories of the policies of its population, and use a Variational Autoencoder (VAE) (Kingma & Welling, 2013) to encode and try to reconstruct the trajectory. The encoder is then used to encode any new policy's trajectory into the latent space of the VAE.

Several other recent works build on this approach and use other information to learn automatically the behavior descriptor from data. RUDA (Grillotti & Cully, 2022b) builds on AURORA but bias the behavior descriptor towards behaviors that are helpful in an adaptation tasks. A similar approach is explored in the multi-agent setting by Dixit et al. (2022), to evolve population of agents where the diversity improves the collaboration in a common task.

In order to validate the interest of those methods, we decided to implement AURORA and to compare its performance with MAP-ELITES in several training and adaptation tasks, to see the potential loss of performance induced by this autonomous way to learn the behavior descriptor. We do not claim any exhaustive study of this method and let this for future work, but want to mention their existence and give an idea of its performance on some of our tasks.

The authors of AURORA were able to provide us a JAX implementation that worked on the ANT robots. To better assess the interest of those methods, we chose a task were no prior was given: ANT-UNI. The data given to the VAE is made of trajectories in the environment, with the full observed state. The proportion of contact time is not given to the algorithm. To the best of our knowledge, AURORA had never been trained or used for adaptation on a UNIDIRECTIONAL task.

We decided to go further and use the unsupervised mechanism of AURORA to handle the diversity of the population of PGA-MAP-ELITES. This variant, that we refer to as PGA-AURORA, is the first attempt to use gradient mechanism inside AURORA.

The results of AURORA and PGA-AURORA are compared to all other methods is reported in Figure 4.

Our results show that the unsupervised behavior discovery method introduced by AURORA resulted in very small performance loss compared to MAP-ELITES and that combining it with PGA-MAP-ELITES even show improved adaptation ability in the whole adaptation range on ANT-UNI with the gravity update and 80% of the adaptation range on the leg dysfunction task.

## J   IMPACT OF THE PRIOR AND OF THE ENVIRONMENT REWARD SIGNAL ON SKILL DISCOVERY METHODS

**Impact of the prior**   The impact of prior in skill discovery methods has already been studied in the literature, for instance in the paper that introduced DADS (Sharma et al., 2019). Nevertheless, we wanted to illustrate this effect on an example taken from one of the tasks of our study, ANT-OMNI.

Our setting is different from the setting used in Sharma et al. (2019) because the algorithms are modified to take into account the reward of the environment. In this task, the reward corresponds to the opposite of the control cost.

To illustrate the effect of the prior, we took the skill discovery methods that performed the best on ANT-OMNI with the prior: DIAYN+REWARD and DADS+REWARD. We removed the prior, and trained the methods again on five seeds. Hence, instead of searching for diversity in the $(x, y)$-space, skills are looking for diversity in the entire state space of the ANT robot.

On Figure 15, we report the evolution of the coverage of all methods along time. On Figure 17, we show the final repertoire and skills of the median obtained for DIAYN+REWARD (resp. DADS+REWARD) in terms of coverage.

The evolution of coverage on Figure 15 shows that the final value is typically divided by four for DADS+REWARD and decreased by 40% for DIAYN+REWARD. The final repertoire and skills, Figure 17, illustrate how consistently the methods are able to navigate in the environment with or without the prior. The impact of the prior would

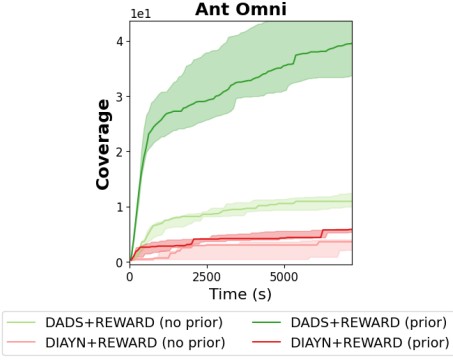

Figure 15: Effect of the prior on skill discovery methods in ANT-OMNI. Evolution of the coverage along time. The given prior is the $(x, y)$ position of the ant's center of gravity.

be smaller in environments were the extrinsic reward can help to explore but it would still have significant impact.

Table 10: Maximum fitness and QD score at the end of the training phase (median over 5 seeds).

|  | ANT-TRAP | ANT-MAZE | ANT-UNI | ANT-OMNI | WALKER-UNI | HALFCHEETAH-UNI |
|---|---|---|---|---|---|---|
| **Maximum fitness** | | | | | | |
| SMERL(DIAYN) | 4.49 ($\pm$0.01)e2 | -7.88 ($\pm$0.41)e3 | **1.71** ($\pm$**0.09**)**e3** | **2.49** ($\pm$**0.00**)**e2** | **1.19** ($\pm$**0.06**)**e3** | 1.89 ($\pm$0.14)e3 |
| SMERL(DADS) | 4.48 ($\pm$0.02)e2 | **-7.62** ($\pm$**0.43**)**e3** | 1.61 ($\pm$0.18)e3 | **2.49** ($\pm$**0.00**)**e2** | 1.17 ($\pm$0.14)e3 | 1.92 ($\pm$0.21)e3 |
| DADS+REWARD | **1.28** ($\pm$**0.41**)**e3** | -7.81 ($\pm$0.49)e3 | 1.24 ($\pm$0.52)e3 | 2.38 ($\pm$0.11)e2 | 6.30 ($\pm$0.48)e2 | 1.89 ($\pm$0.03)e3 |
| DIAYN+REWARD | 9.42 ($\pm$3.43)e2 | -8.13 ($\pm$0.51)e3 | 1.34 ($\pm$0.21)e3 | 2.29 ($\pm$0.04)e2 | 1.00 ($\pm$0.13)e3 | **2.13** ($\pm$**0.27**)**e3** |
| DIAYN | 2.46 ($\pm$0.09)e2 | -1.58 ($\pm$0.06)e4 | 2.19($\pm$0.70)e2 | 2.11 ($\pm$0.24)e2 | 3.42 ($\pm$0.20)e2 | 4.63 ($\pm$0.61)e2 |
| DADS | 2.46 ($\pm$3.84)e2 | -9.16 ($\pm$1.29)e3 | 2.18 ($\pm$0.72)e2 | 2.11 ($\pm$0.68)e2 | 3.50($\pm$0.74)e2 | 4.04 ($\pm$1.43)e2 |
| **QD score** | | | | | | |
| SMERL(DIAYN) | 1.02 ($\pm$0.17)e5 | 7.84 ($\pm$0.44)e5 | 6.20 ($\pm$1.38)e4 | 1.95 ($\pm$1.70)e3 | 3.01 ($\pm$0.20)e5 | 7.92 ($\pm$2.23)e5 |
| SMERL(DADS) | 9.88 ($\pm$0.56)e4 | 8.22 ($\pm$0.07)e5 | 0.83 ($\pm$1.17)e5 | 4.15 ($\pm$1.11)e3 | 2.69 ($\pm$0.38)e5 | 7.51 ($\pm$1.60)e5 |
| DADS+REWARD | **2.42** ($\pm$**0.80**)**e5** | 9.12 ($\pm$0.76)e5 | **2.88** ($\pm$**1.07**)**e5** | 3.05 ($\pm$1.22)e5 | 3.89 ($\pm$0.41)e5 | 8.31 ($\pm$3.91)e5 |
| DIAYN+REWARD | 2.27 ($\pm$0.62)e5 | **9.18** ($\pm$**0.16**)**e5** | 1.29 ($\pm$0.39)e5 | 4.00 ($\pm$0.39)e4 | **4.92** ($\pm$**0.35**)**e5** | **1.68** ($\pm$**0.18**)**e6** |
| DIAYN | 5.80 ($\pm$1.74)e4 | 9.35 ($\pm$2.83)e4 | 7.46 ($\pm$1.29)e4 | 4.84 ($\pm$4.50)e4 | 1.02 ($\pm$0.20)e5 | 4.00 ($\pm$0.50)e5 |
| DADS | 1.37 ($\pm$0.14)e5 | 6.99 ($\pm$2.32)e5 | 1.13 ($\pm$0.29)e5 | **3.50** ($\pm$**0.56**)**e5** | 2.67 ($\pm$0.62)e5 | 7.97 ($\pm$1.13)e5 |

**Impact of the environment reward signal**   Throughout our study, we considered Skill Discovery methods extended to a supervised setting where they take into account the environment reward signal either by summation or with the SMERL method. We advocate that evaluating DADS and DIAYN without this additional supervision would be unfair, since other methods explicitly optimize for the environment's reward. To validate this point, we train DADS and DIAYN in a fully unsupervised setting and measure their performance. In Table 10, we report the maximum fitness and the QD-score obtained by these methods after 2-hours of training. Except on ANT-OMNI (where the environment reward is a penalty for moving), the best score is always attained by a supervised method, in terms of maximum fitness as well as QD-score.

To further emphasize the importance of the environment reward signal, we run adaptation experiments with DADS and DIAYN and evaluate the robustness of the learned policies on perturbed environments. On Figure 16 we report the adaptation results, and compare DADS and DIAYN to their supervised counterparts DIAYN+REWARD, SMERL(DIAYN), DADS+REWARD and SMERL(DADS). It is striking how the performance of DIAYN and DADS plummet without the access to the environment reward signal. It confirms the intuition that without any environement reward signal or finetuning, DADS and DIAYN fail to consistently learn high-performing policies and to adapt in the face of a perturbed environment's dynamics.

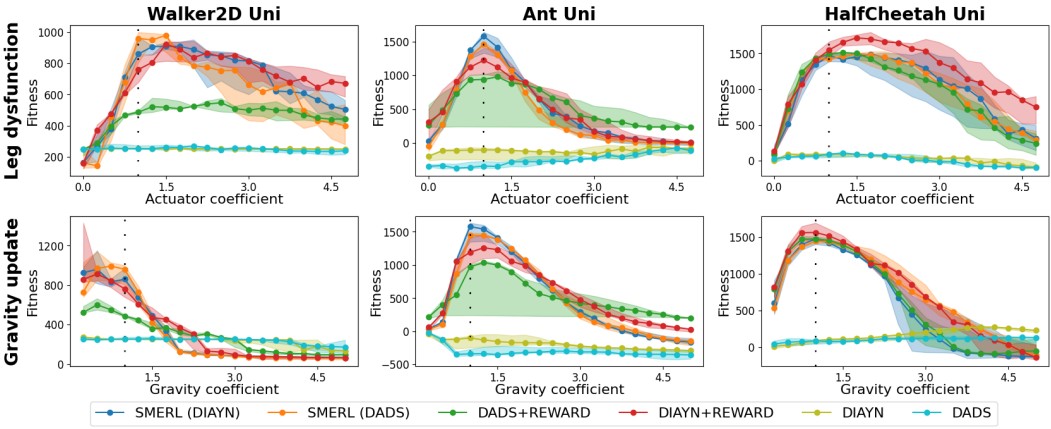

Figure 16: Effect of the environment reward signal on the algorithms DADS and DIAYN on adaptation tasks. Results are the median and interquartile range over 5 random seeds.

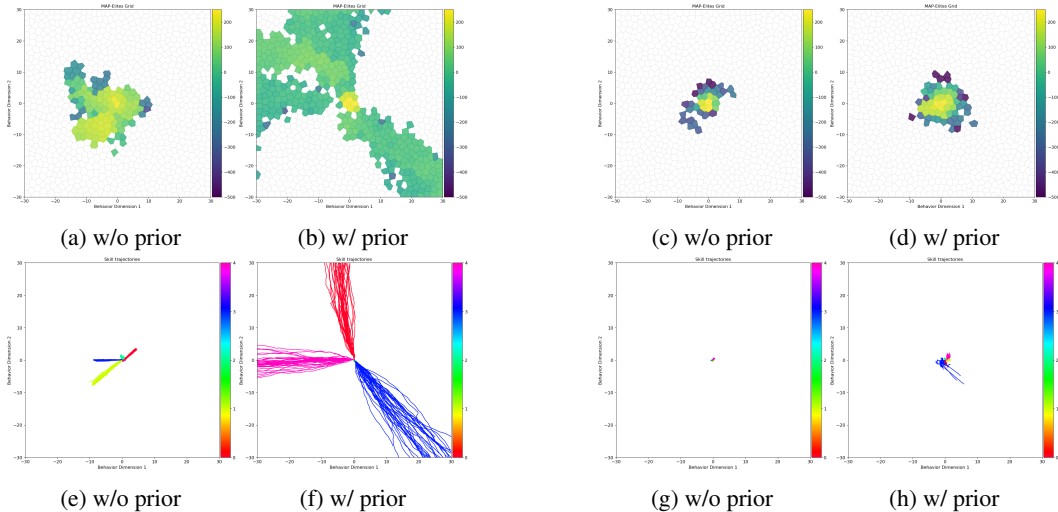

Figure 17: Repertoire and skills obtained by DADS+REWARD (left, (a-b-e-f)) and DIAYN+REWARD (right, (c-d-g-h)) in ANT-OMNI with or without the prior. Using a prior narrows the space where the method is trying to find diversity. When no prior is given, the entire state space is considered. In ANT-OMNI, the prior is the $(x, y)$ position of the ant's center of gravity.

## K   ADDITIONAL STATISTICAL ANALYSIS OF THE RESULTS

In this section, we provide additional data concerning the results in Table 1, Figure 4 and Figure 5.

In Figure 18 we report the performance on various adaptation and hierarchical tasks, similarly to Figure 4 and Figure 5 but using IQM aggregation rather than median aggregation. We observe almost

identical results for all benchmarks, setting aside PGA-AURORA (resp. DIAYN+REWARD) that was ranked third (resp. second) for the ANT-MAZE adaptation task and is now ranked second (resp. third).

In Table 11, we include additional metrics corresponding to mean, median and IQM of the maximum fitness and the QD score over the 5 seeds used in the training phase, whereas Table 1 only includes median for readability purpose. Note that the best method for each environment is almost always the same irrespective of the exact statistic used.

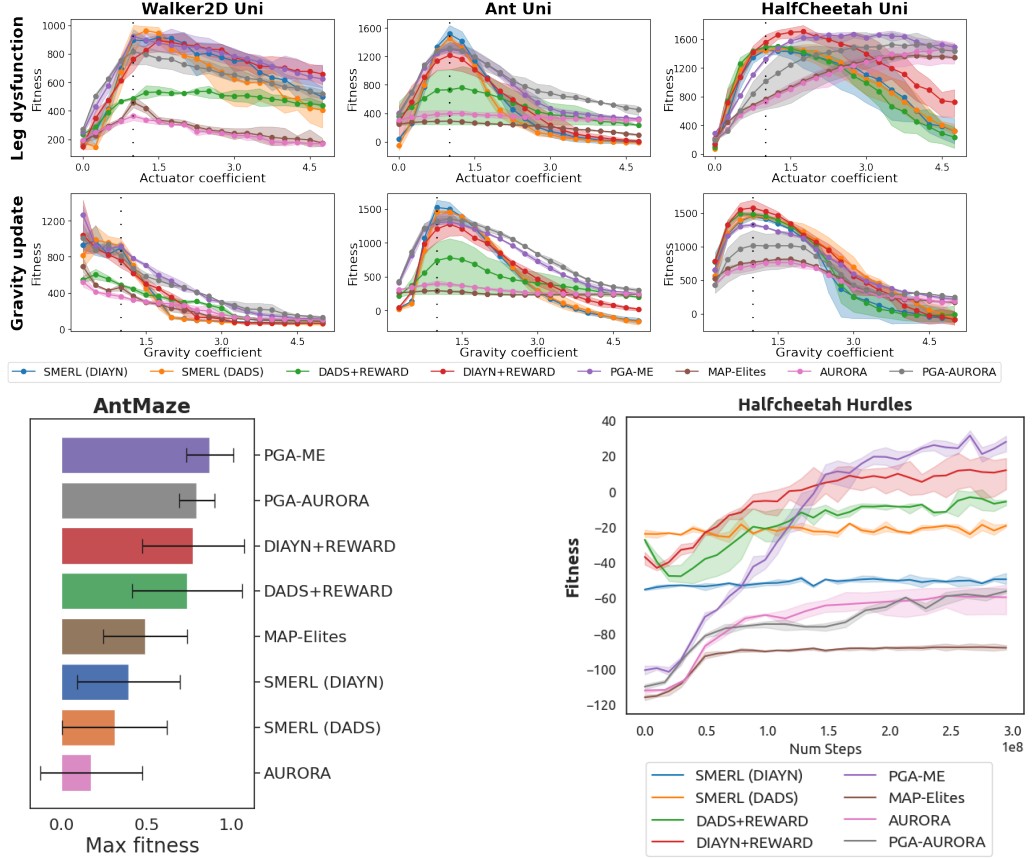

Figure 18: Maximum fitness of the methods when facing a perturbed environment (top) or a modified task (bottom-left) and fitness of the PPO meta-controller for various choices of skill-discovery methods used to generate primitives (bottom-right). IQM aggregation is used instead of median aggregation.

Table 11: Final Max fitness and QD score (training phase). Mean | median | IQM (5 seeds).

|  | ANT-TRAP | ANT-MAZE | POINT-MAZE | ANT-UNI | ANT-OMNI | WALKER-UNI | HALFCHEETAH-UNI |
|---|---|---|---|---|---|---|---|
| **Maximum fitness** | | | | | | | |
| SMERL(DIAYN) | 4.49\|4.49\|4.49 e2 | -8.04\|-7.88\|-7.93 e3 | -5.33\|-3.98\|-5.30 e1 | **1.69\|1.71\|1.69** e3 | 2.49\|2.49\|2.49 e2 | 1.18\|1.19\|1.18 e3 | 1.87\|1.89\|1.91 e3 |
| SMERL(DADS) | 4.48\|4.48\|4.48 e2 | **-7.56\|-7.62\|-7.66** e3 | -5.76\|-8.09\|-6.15 e1 | 1.59\|1.61\|1.60 e3 | 2.49\|2.49\|2.49 e2 | 1.10\|1.17\|1.13 e3 | 1.99\|1.92\|1.96 e3 |
| DADS+REWARD | **1.17\|1.28\|1.30** e3 | -7.65\|-7.81\|-7.78 e3 | -5.76\|-8.09\|-6.15 e1 | 1.13\|1.24\|1.25 e3 | 2.34\|2.38\|2.35 e2 | 6.12\|6.30\|6.12 e2 | 1.89\|1.90\|1.88 e3 |
| DIAYN+REWARD | 7.92\|9.42\|8.14 e2 | -8.34\|-8.13\|-8.33 e3 | -5.52\|-4.48\|-5.65 e1 | 1.46\|1.34\|1.40 e3 | 2.29\|2.29\|2.29 e2 | 9.96\|1.00\|1.01 e3 | **2.05\|2.12\|2.05** e3 |
| PGA-MAP-ELITES | 8.89\|9.77\|9.38 e2 | -9.15\|-9.32\|-9.24 e3 | **-2.26\|-2.26\|-2.25** e1 | 1.46\|1.45\|1.45 e3 | 2.49\|2.49\|2.49 e2 | **1.21\|1.22\|1.23** e3 | 1.74\|1.75\|1.76 e3 |
| PGA-AURORA | 4.70\|4.73\|4.68 e2 | -9.28\|-9.27\|-9.27 e3 | -4.84\|-6.31\|-5.10 e1 | 1.39\|1.44\|1.44 e3 | 2.49\|2.49\|2.49 e2 | 0.99\|1.05\|1.01 e3 | 1.35\|1.47\|1.35 e3 |
| MAP-ELITES | 3.77\|3.81\|3.77 e2 | -1.03\|-1.01\|-1.04 e4 | -2.47\|-2.48\|-2.47 e1 | 4.33\|4.25\|4.27 e2 | 2.49\|2.49\|2.49 e2 | 6.84\|6.88\|6.86 e2 | 1.27\|1.29\|1.27 e3 |
| AURORA | 3.86\|4.06\|4.05 e2 | -1.11\|-1.10\|-1.09 e4 | -2.39\|-2.35\|-2.36 e1 | 5.36\|5.05\|5.26 e2 | 2.49\|2.49\|2.49 e2 | 5.45\|5.71\|5.45 e2 | 1.22\|1.21\|1.19 e3 |
| **QD score** | | | | | | | |
| SMERL(DIAYN) | 1.03\|1.03\|1.06 e5 | 7.69\|7.84\|7.71 e5 | 4.71\|3.89\|4.12 e4 | 6.77\|6.19\|6.66 e4 | 2.52\|1.95\|2.37 e3 | 3.07\|3.01\|3.06 e5 | 7.85\|7.92\|8.13 e5 |
| SMERL(DADS) | 1.01\|0.99\|1.00 e5 | 8.60\|8.22\|8.38 e5 | 8.16\|3.57\|7.15 e3 | 1.25\|8.32\|7.96 e4 | 3.59\|4.15\|3.78 e3 | 2.78\|2.69\|2.67 e5 | 7.90\|7.51\|7.83 e5 |
| DADS+REWARD | 2.24\|2.42\|2.37 e5 | 8.51\|8.22\|8.38 e5 | 4.26\|4.19\|1.12 e4 | 2.55\|2.88\|2.81 e5 | 3.09\|3.03\|3.15 e5 | 3.85\|3.89\|3.83 e5 | 1.01\|0.83\|0.87 e6 |
| DIAYN+REWARD | 2.31\|2.27\|2.25 e5 | 9.22\|9.18\|9.16 e5 | 3.70\|4.04\|3.55 e4 | 1.31\|1.29\|1.36 e5 | 3.92\|4.00\|3.97 e4 | 4.72\|4.92\|4.76 e5 | 1.77\|1.68\|1.73 e6 |
| PGA-MAP-ELITES | **8.23\|7.89\|7.57** e5 | 2.72\|2.74\|2.80 e6 | 3.93\|3.92\|3.93 e5 | **9.19\|9.19\|9.18** e5 | 1.83\|1.58\|1.77 e5 | **8.03\|8.23\|8.11** e5 | 2.99\|2.98\|2.97 e6 |
| PGA-AURORA | 4.41\|4.40\|4.48 e5 | 2.62\|2.59\|2.58 e6 | 2.84\|2.88\|2.75 e5 | 7.34\|7.33\|7.37 e5 | 1.60\|1.69\|1.69 e5 | 5.00\|5.10\|5.07 e5 | 2.40\|2.39\|2.40 e6 |
| MAP-ELITES | 4.74\|4.53\|4.63 e5 | **2.98\|2.98\|2.98** e6 | **4.23\|4.22\|4.22** e5 | 9.04\|9.08\|9.06 e5 | 3.00\|3.05\|3.11 e5 | 6.58\|6.46\|6.58 e5 | **3.05\|3.05\|3.05** e6 |
| AURORA | 3.49\|3.54\|3.78 e5 | 1.85\|2.18\|1.95 e6 | 4.13\|**4.58**\|4.19 e5 | 6.15\|5.82\|6.01 e5 | **3.27\|4.00\|3.44** e5 | 4.46\|4.59\|4.46 e5 | 2.45\|2.45\|2.45 e6 |

## L  ALGORITHMS PSEUDOCODES

We give the pseudocodes for all the algorithms used in our study. The main difference compared to the pseudocodes found in the original papers is that the termination of the algorithm is governed by a time condition. As a reminder, details of our implementations can also be found in Appendix A.1.

---

**Algorithm 1:** MAP-Elites

**Given:**

- Sample size N, time allocation max_time
- MAP-ELITES repertoire $\mathbb{M}$
- $N$ initial policies $\{\pi_{\theta_i}\}_{i=\{1,N\}}$

```
// Main loop
```
current_time $\leftarrow 0$
**while** current_time $<$ max_time **do**

    **if** current_time $> 0$ **then**

```
        // Sampling and mutation
```
        Sample $N$ policies $\{\pi_{\theta_i}\}_{i=1,N}$ from the repertoire $\mathbb{M}$
        Update the parameters $\{\theta_i\}_{i=1,N}$ of the policies with genetic updates

    **end**

```
    // Evaluation and insertion in repertoire
```
    Evaluate fitness and behavior descriptor of the new policies $\{\pi_{\tilde{\theta_i}}\}_{i=1,N}$

    Update current_time

    Add the updated policies in the repertoire $\mathbb{M}$ when relevant

**end**

---

**Algorithm 2:** PGA-MAP-Elites

**Given:**

- Sample size N, time allocation max_time
- MAP-ELITES repertoire $\mathbb{M}$, Replay Buffer $\mathbb{B}$
- $N$ initial policies $\{\pi_{\theta_i}\}_{i=\{1,N\}}$ and a critic $Q_v$

```
// Main loop
```
current_time $\leftarrow 0$
**while** current_time $<$ max_time **do**

    **if** current_time $> 0$ **then**

```
        // Sampling and mutation
```
        Sample N policies $\{\pi_{\theta_i}\}_{i=1,N}$ in repertoire $\mathbb{M}$
        Sample batches of transitions in replay buffer $\mathbb{B}$
        Update half the policies using the critic $Q_v$
        Update the other half with genetic updates

```
        // Train the critic according to TD3 rule
```
        Update the critic $Q_v$

    **end**

```
    // Evaluation and insertion in repertoire
```
    Evaluate the new policies and store collected transitions in buffer $\mathbb{B}$

    Update current_time

    Add the new policies in the repertoire $\mathbb{M}$ when relevant;

**end**

---

---

**Algorithm 3:** AURORA

---

**Given:**

- Sample size $N$, time allocation max_time
- Behavior extraction model $\xi_\theta$, training schedule train_times
- MAP-ELITES repertoire $\mathbb{M}$, $N$ initial policies $\{\pi_{\theta_i}\}_{i=\{1,N\}}$

```
// Main loop
```
current_time $\leftarrow 0$
**while** current_time $<$ max_time **do**

    **if** current_time $> 0$ **then**

```
        // Sampling and mutation
```
        Sample $N$ policies $\{\pi_{\theta_i}\}_{i=1,N}$ from the repertoire $\mathbb{M}$
        Update the parameters $\{\theta_i\}_{i=1,N}$ of the policies with genetic updates
    **end**
```
    // Evaluation and insertion in repertoire
```
    Evaluate fitness and behavior descriptor of the new policies $\{\pi_{\tilde{\theta}_i}\}_{i=1,N}$

    Update current_time

    Add the updated policies in the repertoire $\mathbb{M}$ when relevant

```
    // Training the behavior extraction model
```
    **if** current_time in train_times **then**

        Update $\xi_\theta$ to reconstruct trajectories of policies in $\mathbb{M}$
        Re-evaluate fitness and behavior descriptor of the stored policies
        Update repertoire $\mathbb{M}$ with new behavior descriptors
    **end**
**end**

---

**Algorithm 4:** PGA-AURORA

---

**Given:**

- Sample size $N$, time allocation max_time
- Behavior extraction model $\xi_\theta$, training schedule train_times
- MAP-ELITES repertoire $\mathbb{M}$, $N$ initial policies $\{\pi_{\theta_i}\}_{i=\{1,N\}}$
- Replay Buffer $\mathbb{B}$ and critic $Q_v$

```
// Main loop
```
current_time $\leftarrow 0$
**while** current_time $<$ max_time **do**

    **if** current_time $> 0$ **then**

```
        // Sampling and mutation
```
        Sample $N$ policies $\{\pi_{\theta_i}\}_{i=1,N}$ in repertoire $\mathbb{M}$
        Sample batches of transitions in replay buffer $\mathbb{B}$
        Update half the policies using the critic $Q_v$
        Update the other half with genetic updates

```
        // Train the critic according to TD3 rule
```
        Update the critic $Q_v$
    **end**
```
    // Evaluation and insertion in repertoire
```
    Evaluate the new policies and store collected transitions in $\mathbb{B}$

    Update current_time

    Add the updated policies in the repertoire $\mathbb{M}$ when relevant;

```
    // Training the behavior extraction model
```
    **if** current_time in train_times **then**

        Update $\xi_\theta$ to reconstruct trajectories of policies in $\mathbb{M}$
        Re-evaluate fitness and behavior descriptor of the stored policies
        Update repertoire $\mathbb{M}$ with new behavior descriptors
    **end**
**end**

---

---

**Algorithm 5:** DIAYN+Reward

---

**Given:**

- Time allocation max_time, steps per episode episode_length
- Discriminator model $q_\phi$, Replay Buffer $\mathbb{B}$
- Diversity reward scale $\beta$
- Skill-conditioned policy $\pi_\theta$ and critic $Q_v$, skill prior distribution $p(z)$;

```
// Main loop
current_time ← 0
```
**while** current_time $<$ max_time **do**
    Sample skill $z \sim p(z)$ and initial state $s_0$
    **for** $t \leftarrow 1...$episode_length **do**
```
        // Data collection in the environment
```
        Sample action according to current skill $z$
        Take a step in the environment

```
        // Compute new rewards
```
        Sample batch of transitions from buffer $\mathbb{B}$
        Compute the diversity rewards with the discriminator $r_{\text{diversity}} = \log q_\phi(z|s_{t+1}) - \log p(z)$
        Update reward by combining extrinsic and diversity rewards: $\widetilde{r_t} = r_t + \beta\, r_{\text{diversity}}$

```
        // Update discriminator, policy and critic
```
        Update policy parameters $\theta$ to maximize collected rewards
        Update discriminator parameters $\phi$ to maximize skill/state likelihood
        Update critic parameters $v$ to estimate state/action value
    **end**
**end**

---

**Algorithm 6:** SMERL(DIAYN)

---

**Given:**

- Time allocation max_time, steps per episode episode_length
- Discriminator model $q_\phi$, Replay Buffer $\mathbb{B}$
- Diversity reward scale $\beta$, reward target $R^*$ and threshold $\epsilon$
- Skill-conditioned policy $\pi_\theta$ and critic $Q_v$, skill prior distribution $p(z)$;

```
// Main loop
current_time ← 0
```
**while** current_time $<$ max_time **do**
    Sample skill $z \sim p(z)$ and initial state $s_0$
    **for** $t \leftarrow 1...$episode_length **do**
```
        // Data collection in the environment
```
        Sample action according to current skill $z$
        Take a step in the environment

```
        // Compute new rewards
```
        Sample batch of transitions from buffer $\mathbb{B}$
        Compute the diversity rewards with the discriminator $r_{\text{diversity}} = \log q_\phi(z|s_{t+1}) - \log p(z)$
        Compute sum of extrinsic rewards during the episode associated to each transition $R = \sum_t r_t$
        Compute threshold condition $\delta = \mathbb{1}\!\!\!/_{R>R^*-\epsilon}$
        Update reward by combining extrinsic and diversity rewards: $r_{\text{SMERL}} = r_t + \delta\, \beta\, r_{\text{diversity}}$

```
        // Update discriminator, policy and critic
```
        Update policy parameters $\theta$ to maximize collected rewards
        Update discriminator parameters $\phi$ to maximize skill/state likelihood
        Update critic parameters $v$ to estimate state/action value
    **end**
**end**

---

---

**Algorithm 7:** DADS+Reward

---

**Given:**

- Time allocation max_time, steps per episode episode_length
- Skill-dynamics model $q_\phi$, Replay Buffer $\mathbb{B}$
- Diversity reward scale $\beta$
- Skill-conditioned policy $\pi_\theta$ and critic $Q_v$, skill prior distribution $p(z)$;

```
// Main loop
current_time ← 0
```
**while** current_time < max_time **do**
    Sample skill $z \sim p(z)$ and initial state $s_0$
    **for** $t \leftarrow 1...$episode_length **do**
        `// Data collection in the environment`
        Sample action according to current skill $z$
        Take a step in the environment

        `// Compute new rewards`
        Sample batch of transitions from buffer $\mathbb{B}$
        Compute the diversity rewards with the skill-dynamics $r_{\text{diversity}} = \log(q(s_{t+1}|s_t, z)) - \log(p(s))$
        Update reward by combining extrinsic and diversity rewards: $\widetilde{r_t} = r_t + \beta\, r_{\text{diversity}}$

        `// Update discriminator, policy and critic`
        Update policy parameters $\theta$ to maximize collected rewards
        Update skill-dynamics model parameters $\phi$ to maximize dynamics prediction accuracy
        Update critic parameters $v$ to estimate state/action value
    **end**
**end**

---

**Algorithm 8:** SMERL(DADS)

---

**Given:**

- Time allocation max_time, steps per episode episode_length
- Skill-dynamics model $q_\phi$
- Diversity reward scale $\beta$, reward target $R^*$ and threshold $\epsilon$
- Skill-conditioned policy $\pi_\theta$ and critic $Q_v$, skill prior distribution $p(z)$;

```
// Main loop
current_time ← 0
```
**while** current_time < max_time **do**
    Sample skill $z \sim p(z)$ and initial state $s_0$
    **for** $t \leftarrow 1...$episode_length **do**
        `// Data collection in the environment`
        Sample action according to current skill $z$
        Take a step in the environment

        `// Compute new rewards`
        Sample batch of transitions from buffer $\mathbb{B}$
        Compute the diversity rewards with the skill-dynamics $r_{\text{diversity}} = \log(q(s_{t+1}|s_t, z)) - \log(p(s))$
        Compute sum of extrinsic rewards during the episode associated to each transition $R = \sum_t r_t$
        Compute threshold condition $\delta = \not\Vdash_{R > R^* - \epsilon}$
        Update reward by combining extrinsic and diversity rewards: $r_{\text{SMERL}} = r_t + \delta\, \beta\, r_{\text{diversity}}$

        `// Update discriminator, policy and critic`
        Update policy parameters $\theta$ to maximize collected rewards
        Update skill-dynamics model parameters $\phi$ to maximize dynamics prediction accuracy
        Update critic parameters $v$ to estimate state/action value
    **end**
**end**

---

