# OpenReview forum: "Neuroevolution is a Competitive Alternative to Reinforcement Learning for Skill Discovery"
_ICLR.cc/2023/Conference — ICLR 2023 notable top 25%_

### Official Review · Reviewer_UYjV · 2022-10-19

**Confidence:** 3
**Correctness:** 3
**Technical Novelty And Significance:** 2
**Empirical Novelty And Significance:** 1
**Recommendation:** 5

**Clarity, Quality, Novelty And Reproducibility:**

Clarity: The paper could definitely improve on its formalism of the compared algorithms, with e.g. more mathematical notation. Currently the descriptions of certain methods (e.g. MAP-ELITES) is purely with words and references, which makes it very difficult to know the differences between such algorithms (please see "Weaknesses" above as well).

Quality: The paper appears to be quite rigorous in its experimental study and has multiple ablation studies, both in the main body and ablations.

Novelty: As mentioned above in "Weaknesses", the core issue is that the paper lacks a "punchline" or a compelling purpose, due to its unconvincing analysis.

Reproducibility: The code appears to be quite clean and organized, no issues here.

**Strength And Weaknesses:**

# Strengths
* Comprehensive algorithm list and benchmarking. While my knowledge on the diversity methods is limited, it seems that the provided open source codebase could be a useful tool for future research, due to its extensiveness. From a glance at the Appendix, there also seem to be many ablation studies.
* The overarching goal (i.e. analyze different quality diversity methods and how useful they are) is somewhat solid. The main issue is with its execution, which I provide below.

# Weaknesses
* Lack of a **main punchline**: The current paper acts as an empirical study across previously invented algorithms, rather than inventing an method of its own. This is completely fine, but such a paper's primary focus will then be to provide deep experimental conclusions unknown previously. Currently, the experimental conclusions are not very insightful. A few examples:
    * Section 6: "Our benchmarks however show that no single method outperforms all others, which opens the door to promising work directions" is too vague and can be said of any method in any field.
    * Section 5.2: "We observe that the performance of DADS+REWARD is very sensitive to the exact choice of hyperparameters" - Many RL algorithms are sensitive to hyperparameters; why is this an interesting observation?
    * Section 3: Are any of the methods mentioned here similar or different at a deeper, mathematical level? Currently all of the methods described use mostly words, which makes it hard to determine what components contribute to which behaviors (e.g. without needing to check references, how is MAP-ELITES different from PGA-MAP-ELITES?). It would be far more effective if the paper discussed the mathematical components which are common to certain algorithms, and show experimentally if these components contribute to certain behaviors.

* Factual/Empirical Inaccuracies
    * Section 5.1 "These [standard locomotion] tasks are challenging for evolutionary algorithms that do not leverage backpropagation through the neural policies" - The word "challenging" is too harsh here. ARS [1] has been consistently shown to beat PPO in standard locomotion tasks for a variety of reasons, such as the use of deterministic, linear policies.
    * In Table 4 of the Appendix, I see that a MLP of layer sizes (256, 256) was used. Was this used for all evolutionary methods? The issue is that the performance of evolutionary methods are greatly affected by the problem dimension (i.e. parameter count in this case). It's already been shown that linear policies suffice and are actually sometimes the best [1] for many continuous control tasks. Since the difference in parameter count between a linear policy (order of 100's) and a MLP (20K+) is huge, this can be a very big confounding factor.


[1] Simple random search provides a competitive approach to reinforcement learning (https://arxiv.org/abs/1803.07055)

**Summary Of The Paper:**

At a high level, this paper is an empirical study of a collection of RL algorithms over a collection of environments.
The algorithms are reimplemented versions of:
* Multiple Quality Diversity (QD) methods, which are evolutionary and do not use exact backpropagation.
* Information Theory-Augmented RL methods, which involve adding additional mutual information terms + latent vectors inside a gradient-based RL pipeline.

The environments consist of:
* Unidirectional (Single task)
* Omnidirectional (Multi-task)
* Adaptation
* Additional Exploration (Trap/Maze/etc)

all of which are based on Continuous Control.



**Summary Of The Review:**

I currently lean reject, because of the lack of a compelling purpose or analysis. Currently as written, from a devil's advocate point of view, the paper can easily appear to be a collection of somewhat random algorithms applied on a collection of random benchmarks - the paper needs to have stronger analysis and focus on a compelling purpose for providing these experimental results.

EDIT: I have raised my score to a 5 in light of the authors' rebuttals. I think the writing currently is still not convincing enough to a non-expert such as myself, and I urge the authors to improve on this front.

---

> ### Author Response · Authors · 2022-11-16
> **Response to Reviewer UYjV (part 1/3)**
>
> **This is part 1 out of 3 of the response to reviewer UYjV**
>
> - The overarching goal (i.e. analyze different quality diversity methods and how useful they are) is somewhat solid.
>
> We would like to point out that analyzing different quality diversity methods is not the overarching goal of the paper. The main goal of the paper is to provide an experimental study where we evaluate on an even playing field two seemingly incomparable classes of approaches to skill discovery that have been developed independently by two communities (namely the RL and QD communities) that do not interact much with each other. In this process, we have selected four flagship algorithms from each line of work as representatives and evaluate them extensively in our experimental framework. The selected algorithms all have in common a core component that is a highly distinctive feature of the class of approaches they belong to. Specifically, all considered QD algorithms are built upon the MAP-ELITES algorithm (Mouret & Clune, 2015) while all considered RL algorithms add a diversity term derived from information theory to the reward definition and use state-of-the-art RL algorithms as black boxes. Comparing the performance of individual algorithms is a side contribution that is necessary to identify and highlight general trends about the relative merits of the two classes of approaches, which is really the main focus of this work.
>
> - Lack of a main punchline: The current paper acts as an empirical study across previously invented algorithms, rather than inventing a method of its own. This is completely fine, but such a paper's primary focus will then be to provide deep experimental conclusions unknown previously.
>
> We believe that our work provides three experimental conclusions that were previously unknown:
> 1. QD algorithms attain comparable performance to state-of-the-art skill-discovery RL algorithms on experimental benches typically used to evaluate the latter. To the best of our knowledge, no prior work in the skill-discovery RL literature uses QD methods as baselines for benchmarking purposes, setting aside a small study in the Appendix of Zhou et al. (2022). This is a testament to the fact that this was previously unknown.
> 1. Point 1 is still true even when very limited computational resources are available, provided that the environment simulator is implemented in a modern vectorized framework. Evolutionary methods are known to be trivial to parallelize but they are often seen as being prohibitively slow when limited computational resources are available. In this work, we use the same single affordable hardware accelerator to run the QD and RL algorithms and impose the same time limit to all algorithms (2 hours). In spite of this, QD algorithms attain very competitive results.
> 1. The performance of skill-discovery methods is extremely sensitive to hyperparameters in comparison to QD methods. While RL methods are broadly known to be sensitive to hyperparameters, there are very few experimental studies quantifying the impact this can have on performance in comparison to other methods. Given that hyperparameter tuning can be computationally expensive, it is all the more important to give concrete data points to readers that may not be familiar with either line of work.
>
> - Section 6: "Our benchmarks however show that no single method outperforms all others, which opens the door to promising work directions" is too vague and can be said of any method in any field.
>
> We agree that the wording was too vague and we have adjusted the formulation in the paper. We meant to say that skill-discovery RL algorithms do not outperform QD algorithms on skill-discovery experimental benches and vice-versa, which suggests that hybridization of methods may be a promising - yet largely unexplored - research direction.  We believe that this was largely unknown to both the RL and QD communities. The fact that, setting aside a small study in the appendix of (Zhou et al., 2022), no prior work in the skill-discovery RL literature uses QD methods as baselines for benchmarking purposes is a case in point.
>
> - Section 5.2: "We observe that the performance of DADS+REWARD is very sensitive to the exact choice of hyperparameters" - Many RL algorithms are sensitive to hyperparameters; why is this an interesting observation?
>
> The interesting observation, substantiated in the last paragraph of Section 5.2, is that DADS+REWARD is much more sensitive to the exact choice of hyperparameters than MAP-ELITES. Specifically, our study suggests that a skill-discovery RL method will likely require a more thorough hyperparameter tuning search than a QD method to achieve the same level of performance. This needs to be accounted for when contemplating methods given the computational cost of hyperparameter tuning.

---

> ### Author Response · Authors · 2022-11-16
> **Response to Reviewer UYjV (part 2/3)**
>
> **This is part 2 out of 3 of the response to reviewer UYjV**
>
> - Section 3: Are any of the methods mentioned here similar or different at a deeper, mathematical level? Currently all of the methods described use mostly words, which makes it hard to determine what components contribute to which behaviors (e.g. without needing to check references, how is MAP-ELITES different from PGA-MAP-ELITES?). It would be far more effective if the paper discussed the mathematical components which are common to certain algorithms, and show experimentally if these components contribute to certain behaviors.
>
> We have added pseudocodes for all the algorithms under study in Section L of the Appendix to make it easier to identify patterns and spot differences. The consistent use of notations across pseudocodes to refer to the same components should also help in this respect. Additionally, we have included a new table in the Appendix (namely Table 2 in Section A) that enables the reader to identify at a glance which components are shared across algorithms. Note that all QD algorithms are built upon MAP-ELITES while all considered skill-discovery RL algorithms add a diversity term derived from information theory to the reward definition and use state-of-the-art RL algorithms as black boxes. In this work, our goal is to highlight general trends about the relative merits of these two classes of approaches rather than identifying algorithm-specific components that contribute to specific behaviors. The latter is out of scope for this study as this would have required a significant number of additional experiments on top of the ones already included in the paper to be able to draw meaningful conclusions.
>
> - How is MAP-ELITES different from PGA-MAP-ELITES?
>
> The pseudocodes added in the Appendix (Section L) should make it easier to identify differences between these two algorithms. The difference between MAP-ELITES and PGA-MAP-ELITES comes down to the alternate use of policy gradients to update the policies. In MAP-ELITES, policies are always updated using genetic mutations. In PGA-MAP-ELITES, a single critic is trained alongside a dedicated actor in a similar fashion to what is done in the TD3 algorithm using the experience collected during the evaluations of policies. Policies sampled from the repertoire are updated using either genetic mutations or policy gradients which are themselves derived using the critic.
>
> - Section 5.1 "These [standard locomotion] tasks are challenging for evolutionary algorithms that do not leverage backpropagation through the neural policies" - The word "challenging" is too harsh here. ARS [1] has been consistently shown to beat PPO in standard locomotion tasks for a variety of reasons, such as the use of deterministic, linear policies.
>
> First, the performances, in terms of maximum episodic reward achieved, reported in [1] for standard MuJoCo locomotion tasks are still far below what can be achieved with slightly tuned versions of standard offline RL algorithms such as SAC and TD3, compare Figure 2 of [1] with Figure 9 of Ota et al. (2021).
>
> Second, some evolutionary algorithms like ES (Such et al., 2018; Colas et al., 2020) are shown to be able to evolve high-fitness policies in such environments but they require several orders of magnitude more environment steps than state-of-the-art RL off-policy algorithms (TD3, SAC) to evolve these policies. By “challenging”, we meant to say that evolutionary algorithms either require a significant number of environment interactions to solve these tasks or completely fail. We have updated the formulation in the paper since it was ambiguous.

---

> ### Author Response · Authors · 2022-11-16
> **Response to Reviewer UYjV (part 3/3)**
>
> **This is part 3 out of 3 of the response to reviewer UYjV**
>
> - In Table 4 of the Appendix, I see that a MLP of layer sizes (256, 256) was used. Was this used for all evolutionary methods? The issue is that the performance of evolutionary methods are greatly affected by the problem dimension (i.e. parameter count in this case). It's already been shown that linear policies suffice and are actually sometimes the best [1] for many continuous control tasks. Since the difference in parameter count between a linear policy (order of 100's) and a MLP (20K+) is huge, this can be a very big confounding factor.
>
> In order to keep the comparison relatively fair, we decided to use the same neural network architecture and the same number of parameters for all methods. Additionally, we use a number of parameters close to what is typically used in prior skill-discovery RL works ((256, 256) for SMERL, (300, 300) for DIAYN, and between (128, 128) and (1024, 1024) for DADS). While this may disadvantage QD methods that rely exclusively on genetic mutations, i.e. MAP-ELITES and AURORA in our study, this is not necessarily the case for PGA-MAP-ELITES and PGA-AURORA. In fact, the authors of the original study introducing PGA-MAP-ELITES use layer sizes of (128, 128).
>
> Another option would have been to tune the number of parameters for each algorithm individually but this would have required a very large number of additional runs as we would have had to do this in a systematic fashion for all algorithms since SAC and TD3 (which are used internally by all skill-discovery RL methods considered in this work as well as PGA-MAP-ELITES and PGA-AURORA) are also known to be able to benefit from wider networks, see Ota et al. (2021). However, note that it is unclear if these methods would benefit from wider networks in our study given the extra computational cost this incurs and the limited time budget.
>
> - Currently as written, from a devil's advocate point of view, the paper can easily appear to be a collection of somewhat random algorithms applied on a collection of random benchmarks
>
> Paraphrasing our first answer: we have selected four flagship algorithms from both the skill-discovery RL literature and the QD literature (so eight algorithms in total). The selected algorithms all have in common a core component that is a highly distinctive feature of the class of approaches they belong to. Specifically, all considered QD algorithms are built upon the MAP-ELITES algorithm (Mouret & Clune, 2015) while all considered RL algorithms add a diversity term derived from information theory to the reward definition and use state-of-the-art RL algorithms as black boxes. The four QD algorithms we have selected cover a wide range of popular approaches in the QD literature: evolutionary-only approaches with MAP-ELITES, approaches combining evolutionary techniques with gradient-based ones with PGA-MAP-ELITES, unsupervised evolutionary-only approaches with AURORA, and a combination of the last two with PGA-AURORA. Conversely, the skill-discovery RL algorithms we have selected are breakthrough papers in this nascent field with high citation counts.
>
> As for the benchmarks, we believe that we cover a large spectrum of the environments and tasks that are used in the skill-discovery RL literature and the QD literature to evaluate algorithms on their abilities to discover skills.
>
>
> We thank the reviewer for the detailed and relevant review as well as for suggesting improvements.
>
> Ota, K., Jha, D. K., & Kanezaki, A. (2021). Training larger networks for deep reinforcement learning. arXiv preprint arXiv:2102.07920.
>
> Such, Felipe Petroski et al. “Deep Neuroevolution: Genetic Algorithms Are a Competitive Alternative for Training Deep Neural Networks for Reinforcement Learning.” ArXiv abs/1712.06567 (2017): n. pag.
>
> Cédric Colas, Vashisht Madhavan, Joost Huizinga, and Jeff Clune. 2020. Scaling MAP-Elites to deep neuroevolution. In Proceedings of the 2020 Genetic and Evolutionary Computation Conference (GECCO '20). Association for Computing Machinery, New York, NY, USA, 67–75. https://doi.org/10.1145/3377930.3390217

---

### Official Review · Reviewer_yJjm · 2022-10-24

**Confidence:** 3
**Correctness:** 4
**Technical Novelty And Significance:** 2
**Empirical Novelty And Significance:** 4
**Recommendation:** 6

**Clarity, Quality, Novelty And Reproducibility:**

The paper is clearly written, although in parts a little more context/details would be good. Of course this is hard to balance with the page limit.

The empirical comparison seems quite thorough, and I am not aware of a similar comparison in the literature. This should be of interest to the ICLR community.

Reproducing this work would be a substantial undertaking but should be achievable with the submitted code.

**Strength And Weaknesses:**

Strengths:
* extensive careful empirical evaluation with interesting results

Weaknesses:
* comparing with the same time budget introduces dependencies on hardware, and implementation quality that are difficult to assess. I would at least report roughly how many environment interactions each methods processes to give readers an indication
- Related to the first point I’m unclear why RL methods are less able to leverage accelerators (i.e. more env interactions). Perhaps the authors could provide their intuitions on this point. Is this because recent methods have been developed/tuned in a different regime?
- Unclear if any of the methods in this paper could scale to something that seems less toy (e.g. dexterous manipulation with a robotic hand or humanoid locomotion)
- I appreciate that the paper is has a lot results but having a limited understanding of QD methods I might have benefitted from more details.
- Consider citing Hausman et al. (https://openreview.net/forum?id=rk07ZXZRb) which is pretty similar to DIAYN + reward (and predates DIAYN).
- Figures are all a bit small and hard to read.


**Summary Of The Paper:**

This paper compares RL methods with information theory inspired diversity rewards and quality diversity methods for skill discovery on a large set of environments implemented in jax/brax (and thus able to take advantage of accelerators). It compares methods in terms of diversity metrics, performance on adaptation tasks and usefulness for hierarchical planning and finds that QD methods are competitive and sometimes perform better and potentially less sensitive to hyperparameters.

**Summary Of The Review:**

Nice paper presenting an empirical comparison between RL-based skill discovery methods and QD methods. I think the paper can be improved in some ways but presents interesting empirical results.

---

> ### Author Response · Authors · 2022-11-16
> **Reponse to Reviewer yJjm (part 1/2)**
>
> **This is part 1 out of 2 of the response to reviewer yJjm**
>
> - Comparing with the same time budget introduces dependencies on hardware, and implementation quality that are difficult to assess. I would at least report roughly how many environment interactions each methods processes to give readers an indication
>
> Note that this information was already included in Section C of the Appendix in the form of a table (namely Table 8) showing the total number of environment steps (averaged over 5 seeds) carried out per algorithm over the course of training. The evolution of the QD score and fitness as a function of the number of environment steps is also reported in this section of the Appendix for all methods.
>
> We agree that a comparison based on a fixed time budget introduces dependencies on hardware. However, we believe that runtime considerations are as important as sample efficiency for a practitioner whenever a faithful simulator is available. QD methods are able to perform one or two orders of magnitude more environment steps than MI-based methods in the same amount of time.
>
> Regarding the quality of the implementation: we agree that the implementation quality is difficult to assess. Nevertheless, for both classes of methods, we did our best to optimize the implementations in our experiments. For QD methods, we based our implementations on those of QDax (Lim et al., 2022), that were reported to be several orders of magnitude faster than previous MAP-Elites implementations. For MI RL methods, taking the example of DIAYN, we perform on average 1e7 gradient updates (one per environment step) in the course of training, while the authors of DIAYN report at most 1e6 gradient steps (one per environment step). We also want to point out that we ran all our experiments on the same hardware, an affordable GPU (NVIDIA Quadro RTX 4000), for all methods.
>
> - Related to the first point I’m unclear why RL methods are less able to leverage accelerators (i.e. more env interactions). Perhaps the authors could provide their intuitions on this point. Is this because recent methods have been developed/tuned in a different regime?
>
> QD methods are usually less data-efficient than RL methods and require more environment interactions to converge. As a consequence, anything that speeds up environment interactions immediately benefits QD methods more than RL ones. Conversely, off-policy RL algorithms cannot get the same speed-up as their bottleneck lies in the back-propagation updates: as explained by the authors of Brax (a JAX-based framework for environment simulations) regarding their SAC implementation “we observed that the training throughput now becomes bottlenecked by SGD updates (12% for running the env, 10% for working with replay buffer, 78% for SGD updates)”, see Freeman et al. (2021).
>
> - Unclear if any of the methods in this paper could scale to something that seems less toy (e.g. dexterous manipulation with a robotic hand or humanoid locomotion
>
> We agree with the reviewer that it is desirable to benchmark methods on complex problems. However, we want to stress that the environments we used, such as Ant, Halfcheetah, and Walker, have been largely used in (i) the skill-discovery literature to demonstrate algorithms’ abilities to learn locomotion primitives (Eysenbach et al., 2018; Sharma et al., 2019; Gaya et al., 2022) and (ii) the continuous control literature to benchmark the performance of algorithms (Haarnoja et al., 2018; Schulman et al., 2017; Fujimoto et al., 2018). Furthermore, previous works in the QD literature have consistently used the environments we selected as hard control-exploration problems, namely AntMaze and AntTrap (Pierrot et al. 2022), and for the learning of diverse locomotion gaits, namely the Uni environments (Nilsson et al., 2021; Cully et al., 2015).
>
> That being said, it is true that the more the environments the better in a benchmark. However, we want to point out that we already evaluate 8 algorithms on 7 environments for a total of 56 algorithm-environment pairs (each pair being trained with 5 random seeds). For 32 of these pairs, around 30 adaptation/hierarchical settings were evaluated for a total of about a thousand settings (each setting being repeated with 5 random seeds).
>
> - I appreciate that the paper has a lot of results but having a limited understanding of QD methods I might have benefitted from more details.
>
> Note that additional details are provided for all algorithms under study in Section A of the Appendix. Additionally, we have now added pseudocodes for all of them in Section L of the Appendix to make the paper more self-contained. The consistent use of notations across pseudocodes to refer to the same components should also help identify patterns.
>
> - Consider citing Hausman et al. (https://openreview.net/forum?id=rk07ZXZRb) which is pretty similar to DIAYN + reward (and predates DIAYN).
>
> Thank you for the relevant missing reference. We have added it in the paper.

---

> ### Author Response · Authors · 2022-11-16
> **Reponse to Reviewer yJjm (part 2/2)**
>
> **This is part 2 out of 2 of the response to reviewer yJjm**
>
> - Figures are all a bit small and hard to read.
>
> Thank you for pointing this out. This was true especially for Figure 2 and 4, we have increased the font size and tried to reduce the white spaces as much as possible in the revision.
>
> We thank the reviewer for the detailed review and for suggesting improvements. We have updated the revision to address your comments. We are thankful for you pointing out the fact that our comparison is thorough and could benefit the ICLR community. We hope it will inspire follow-up works mixing QD and MI-based methods.
>
> Eysenbach B, Gupta A, Ibarz J, Levine S. Diversity is all you need: Learning skills without a reward function. arXiv preprint arXiv:1802.06070. 2018 Feb 16.
>
> Sharma A, Gu S, Levine S, Kumar V, Hausman K. Dynamics-aware unsupervised discovery of skills. arXiv preprint arXiv:1907.01657. 2019 Jul 2.
>
> Gaya, J., Soulier, L., & Denoyer, L. (2022). Learning a subspace of policies for online adaptation in Reinforcement Learning. ArXiv, abs/2110.05169.
>
> Haarnoja T, Zhou A, Hartikainen K, Tucker G, Ha S, Tan J, Kumar V, Zhu H, Gupta A, Abbeel P, Levine S. Soft actor-critic algorithms and applications. arXiv preprint arXiv:1812.05905. 2018 Dec 13.
>
> Schulman J, Wolski F, Dhariwal P, Radford A, Klimov O. Proximal policy optimization algorithms. arXiv preprint arXiv:1707.06347. 2017 Jul 20.
>
> Fujimoto S, Hoof H, Meger D. Addressing function approximation error in actor-critic methods. InInternational conference on machine learning 2018 Jul 3 (pp. 1587-1596). PMLR.
>
> Thomas Pierrot, Valentin Macé, Felix Chalumeau, Arthur Flajolet, Geoffrey Cideron, Karim Beguir, Antoine Cully, Olivier Sigaud, and Nicolas Perrin-Gilbert. 2022. Diversity policy gradient for sample efficient quality-diversity optimization. In Proceedings of the Genetic and Evolutionary Computation Conference (GECCO '22). Association for Computing Machinery, New York, NY, USA, 1075–1083. https://doi.org/10.1145/3512290.3528845
>
> Nilsson O, Cully A. Policy gradient assisted map-elites. In Proceedings of the Genetic and Evolutionary Computation Conference 2021 Jun 26 (pp. 866-875).
>
> Cully A, Clune J, Tarapore D, Mouret JB. Robots that can adapt like animals. Nature. 2015 May;521(7553):503-7.
>
> Freeman CD, Frey E, Raichuk A, Girgin S, Mordatch I, Bachem O. Brax--A Differentiable Physics Engine for Large Scale Rigid Body Simulation. arXiv preprint arXiv:2106.13281. 2021 Jun 24.

---

### Official Review · Reviewer_rzaa · 2022-10-24

**Confidence:** 3
**Correctness:** 4
**Technical Novelty And Significance:** 2
**Empirical Novelty And Significance:** 3
**Recommendation:** 8

**Clarity, Quality, Novelty And Reproducibility:**

The paper is clearly written and provides an interesting unified perspective on how both lines of research intersect.

**Strength And Weaknesses:**

Strengths:This paper provides a comprehensive and important benchmark connecting RL based skill discovery with Quality Diversity approaches. It provides three settings for comparison using metrics, using few shot generalisation on adaptive environments, and using primitives for hierarchical planning.
Weaknesses:
The metrics provided in Table 1 and figure 2 seem to be biased towards QD methods since RL based on MI does not naturally have or optimise for QD score, can the authors provide additional metrics that are perhaps less bias?
In the hierarchical learning experiments can the authors clarify the number of seeds used? And are there other environments that can be used for this setting?


**Summary Of The Paper:**

This paper provides a comparison and benchmark of Evolutionary strategies and rl based skill discovery approaches. The authors provide a thorough literature review on the most recent approach in both lines of research and compare them in light of different metrics, how they generalise to changing environments (adaptive gravity, motor failure and different target locations), and how they can be use in hierarchical planning.

**Summary Of The Review:**

I recommend this paper to be accepted since it provides a comparison of approaches in MI RL and quality diversity approaches for skill discovery. They empirically analyse how different methods on each side fare in terms of metrics for diversity and in terms of few shot generalisation.

---

> ### Author Response · Authors · 2022-11-16
> **Response to Reviewer rzaa**
>
> - The metrics provided in Table 1 and figure 2 seem to be biased towards QD methods since RL based on MI does not naturally have or optimise for QD score, can the authors provide additional metrics that are perhaps less bias?
>
> We agree with the reviewer that measuring diversity using metrics from the QD literature creates a bias in favor of QD methods and this was already highlighted in the main body of the paper. However, as detailed in Section 5.2, we strived to mitigate the bias by 1. giving priors to skill-discovery RL methods that make use of the definitions of the behavior descriptor spaces whenever possible and 2. using QD methods, namely AURORA and PGA-AURORA, that are not directly provided with the definitions of the behavior descriptor spaces.
>
> Finally, we want to point out that the skill-discovery field is still nascent and that there is no consensus yet on how to assess the skills’ quality in the community. This is why, following previous works, we also evaluated the skill-discovery methods on the basis of their performances on downstream tasks (few-shot adaptation tasks and hierarchical learning tasks). That said, we believe that the comparison based on metrics derived from the QD literature is insightful because it shows that MI-RL-based methods actually also perform generally quite well on that front.
>
> - In the hierarchical learning experiments can the authors clarify the number of seeds used?
>
> Thank you for pointing out this omission. We use 5 random seeds for the hierarchical learning experiments. We have added this missing piece of information in the caption of Figure 5.
>
> - And are there other environments that can be used for this setting?
>
> We would like to point out that there is no de-facto standard hierarchical RL environments in the skill-discovery literature. Among the existing environments, we selected Halfcheetah Hurdles because it requires the meta-controller to combine different locomotion gaits to get through the hurdles, which certainly rewards efficient and diverse populations of solutions. Given space constraints, it was hard to include additional experiments in the paper (it already took quite an effort to make the ones currently included in the paper fit in 9 pages).
>
>
> Thank you very much for the thorough review. We hope that we have answered your questions properly. We are grateful for your comments about the public utility of this work. We do believe it could benefit both the RL and QD community and encourage further unifying works.

---

> > ### Comment · Reviewer_rzaa · 2022-11-17
> > **thank you for the comments**
> >
> > most of my concerns have been addressed. I would like to thank the authors for the answers provided.

---

### Official Review · Reviewer_hXTi · 2022-10-25

**Confidence:** 4
**Correctness:** 4
**Technical Novelty And Significance:** 3
**Empirical Novelty And Significance:** 3
**Recommendation:** 8

**Clarity, Quality, Novelty And Reproducibility:**

Clarity: High. The paper is clear and easy to follow. The prior and related work is properly cited and helps the reader to understand the manuscript faster.

Quality: High. The paper includes extensive experiments and a thorough analysis of prior work as well as new techniques that use QD for skill discovery.

Novelty: High. While QD and skill discovery are mature fields at this point, this paper is the first attempt for using QD for skill discovery, as far as I’m concerned.

Reproducibility: High. The code is included alongside the submission. I have no worries regarding the reproducibility.

**Strength And Weaknesses:**

## Strengths

- The paper takes on an exciting direction, using QD for skill discovery. While the MI-RL line of work has shown great results in recent years, QD methods can provide an alternative, given the diversity element that is being explicitly optimized for.
- I like how the paper positions itself neutrally to both types of approaches, namely QD and MI-RL. After presenting all the experimental results, it comes to the conclusion that no method is optimal for all environments and lays down avenues for future work. It also accepts the pros and cons of both approaches, and analyses all methods without any biases, as far as I could tell.
- The paper includes extensive empirical evaluations and analysis, including experiments directly evaluating diversity, few-shot adaption and hierarchical learning. The list of baselines used in the experiments is fair, as far as I’m concerned.

## Weaknesses

- One weakness of the work is that it always provides the median of the metric of interest over the seeds (e.g. Table 1, Figure 4). This can provide unfair results if there are outliers. I suggest that the authors provide several metrics of interest, such as mean, IQM, etc. [rliable]([https://github.com/google-research/rliable](https://github.com/google-research/rliable)) is a great tool for this.
- I didn’t quite understand why haven’t the authors include rewardless DIAYN and DADS results in their experiments. Throughout Section 5, they only use DIAYN/DADS+REWARDS as baselines.
- QD-based methods are obviously more expensive computationally, given they store separate weights for each policy/skill. The authors argue that this is okay if the simulator is very fast. I have no arguments against it, but it would be great to know how much more compute-heavy QD-based methods are. The authors only mention that everything is within 2 hours of training, but there can be a big difference within this range.
- Minor: please use the correct citation styles of prior work. See the ICLR instruction template for information on where to use \citep and \citet.

Note: My score is not final and can change (in either direction) based on the authors' responses or comments of other reviewers.

# Post rebuttal update

I have increased my score based on the responses from the authors.

**Summary Of The Paper:**

This paper proposes and integrates the usage of Quality-Diversity methods for skill discovery in RL. While prior methods largely rely on maximum-information (MI) RL, this paper makes a case for QD as an alternative approach when used with fast environment simulators. The authors provide a large number of experiments which show that QD-based methods are competitive with the MI RL approach and should be researched in more dept as future work.

**Summary Of The Review:**

Interesting work that combines QD and skill discovery in RL. Extensive experiments and thorough analysis of prior work are included in the manuscript. I have raised several questions and some concerns that I’d like to be answered by the authors during the rebuttal.

---

> ### Author Response · Authors · 2022-11-16
> **Response to Reviewer hXTi**
>
> - One weakness of the work is that it always provides the median of the metric of interest over the seeds (e.g. Table 1, Figure 4). This can provide unfair results if there are outliers. I suggest that the authors provide several metrics of interest, such as mean, IQM, etc. rliable is a great tool for this.
>
> Thank you for pointing this out. In addition to the standard deviation that was already reported in Table 9 in the Appendix, we now report the mean and IQM of the maximum fitness and QD score for all methods under study in Table 10 in the Appendix. Moreover, we also report the performance w.r.t. metrics of interest using the IQM statistic for the few-shot adaptation experiments and the hierarchical learning experiments in Figure 20, instead of the median statistic originally used for Figures 4 and 5. Overall, we draw the same conclusions about the relative merits of QD methods and skill-discovery RL methods using either statistic, though the rankings of individual algorithms vary slightly for each set of experiments depending on the exact statistic used.
>
>
> - I didn’t quite understand why haven’t the authors include rewardless DIAYN and DADS results in their experiments. Throughout Section 5, they only use DIAYN/DADS+REWARDS as baselines.
>
> The performance of rewardless DIAYN and DADS was reported in Section J of the Appendix for the experiments directly evaluating diversity and the few-shot adaptation experiments. Our results show that these methods are consistently outperformed by the variations considered in the main body of the paper that explicitly use the extrinsic rewards (namely DIAYN/DADS+Reward). As a result, we decided to defer these results to the Appendix. For the sake of clarity, we now explain this choice in the main body of the paper.
>
> - QD-based methods are obviously more expensive computationally, given they store separate weights for each policy/skill. I have no arguments against it, but it would be great to know how much more compute-heavy QD-based methods are.
>
> Thank you for pointing out the lack of details about this aspect of the comparison. To be more precise, the fact that QD methods store separate weights for each policy makes them memory intensive rather than computationally expensive. That said, the memory usage of QD methods was not a bottleneck in our study, even though we were working with a single affordable GPU (a Quadro RTX 4000). Storing all 1024 policies (with 2x256 hidden layers) takes 0.6 GB, which is only a fraction of the 8GB of memory available on these GPUs.
>
> - The authors argue that this is okay if the simulator is very fast.
>
> A well-documented limitation of QD methods (and more generally evolutionary methods) lies in their relatively poor data efficiency: these methods tend to extract less information from a given number of environment interactions than RL methods do, hence they typically require more interactions. Several frameworks (including Jax) make it possible to automatically vectorize computations and to leverage hardware accelerators for environment simulations which can speed up QD methods by multiple orders of magnitude (Lim et al., 2022). To get a better idea of the data-efficiency and time-efficiency of the methods under study, we provide the evolution of metrics of interest during training as a function of both the total number of environment interactions and the runtime for all methods in Section C of the Appendix.
>
> - The authors only mention that everything is within 2 hours of training, but there can be a big difference within this range.
>
> We train each method for exactly two hours. We have updated the wording in the paper given that it was a bit ambiguous. Note that the evolution of metrics of interest during training are also reported in Section C of the Appendix for all methods to get a sense of the dynamics of training.
>
> - Minor: please use the correct citation styles of prior work. See the ICLR instruction template for information on where to use \citep and \citet.
>
> Thank you for pointing this out. We have updated the citation style everywhere.
>
> - Interesting work that combines QD and skill discovery in RL. Extensive experiments and thorough analysis of prior work are included in the manuscript. I have raised several questions and some concerns that I’d like to be answered by the authors during the rebuttal.
>
> Thank you very much for your review and all your insightful remarks. We have updated the paper accordingly. We really appreciate the fact that you value the neutrality of our study as well as our effort to reduce the bias. We really hope this will benefit the research community.
>
> Bryan Lim, Maxime Allard, Luca Grillotti, and Cully Antoine. Accelerated quality-diversity for
> robotics through massive parallelism. CoRR, abs/2202.01258, 2022. URL http://arxiv.
> org/abs/2202.01258.

---

> > ### Comment · Reviewer_hXTi · 2022-11-18
> > **Thank you for your response**
> >
> > I thank the authors for their response. I thank them for including additional results that I have asked for addressing some of my concerns. I have no follow-up questions at this stage and have raised my score..

---

### Author Response · Authors · 2022-11-16
**General response to reviewers**

First we would like to thank the reviewers for their suggestions that will surely improve the paper. We answer each reviewer independently point by point.

---

### Decision · Program_Chairs · 2023-01-20

**Decision:**

Accept: notable-top-25%

**Justification For Why Not Higher Score:**

The paper does not necessarily have a groundbreaking result nor it has a clear take-home message. I would expect that for a higher score. The main virtue of the paper is a careful empirical analysis of two types of methods.

**Justification For Why Not Lower Score:**

All reviewers agreed upon the value of the paper, even the reviewer who gave it weak reject acknowledged that the paper makes interesting contributions but the writing should be improved, which can be done in the camera-ready version of the paper.

I'm recommending a spotlight talk because it bridges different fields in ML and the paper will benefit from additional visibility.

**Metareview: Summary, Strengths And Weaknesses:**

All reviewers agreed that this comprehensive empirical study comparing two classes of methods (information-theory-inspired skill discovery in RL and quality diversity from the neuroevolution community) is very valuable to the field. This is relevant not only in terms of performance itself but also in terms of defining a common ground, cross-evaluation across environments designed for both types of methods, and so on.

There is already enough empirical data in this paper, so no additional experiments are needed. That being said, reviewers pointed out the writing could be improved. Multiple recommendations were given, such as more concisely summarizing the take-home message of the paper. I urge the authors to go over the reviewers’ recommendations and incorporate as much as you can in the camera ready version of the paper.


**Note From Pc:**

if the above contains the word "oral" or "spotlight" please see: "oral" presentation means -> notable-top-5% and "spotlight" means -> notable-top-25%. As stated in our emails, we are disassociating presentation type from AC recommendations

**Summary Of Ac-Reviewer Meeting:**

Not applicable.